# Targeted metagenomics reveals association between severity and pathogen co-detection in infants with respiratory syncytial virus

Gu-Lung Lin [1,2] ✉, Simon B. Drysdale [1,2,3], Matthew D. Snape[1,2], Daniel O'Connor [1,2], Anthony Brown [4], George MacIntyre-Cockett[5], Esther Mellado-Gomez[5,6], Mariateresa de Cesare[5,7], M. Azim Ansari[4,5], David Bonsall[5,8], James E. Bray [9], Keith A. Jolley [9], Rory Bowden [5,10,11], Jeroen Aerssens[12], Louis Bont [13,14], Peter J. M. Openshaw [15], Federico Martinon-Torres [16,17,18], Harish Nair [19,20], Tanya Golubchik [8,21,22] ✉, Andrew J. Pollard [1,2,22] & RESCEU Consortium*

Respiratory syncytial virus (RSV) is the leading cause of hospitalisation for respiratory infection in young children. RSV disease severity is known to be age-dependent and highest in young infants, but other correlates of severity, particularly the presence of additional respiratory pathogens, are less well understood. In this study, nasopharyngeal swabs were collected from two cohorts of RSV-positive infants <12 months in Spain, the UK, and the Netherlands during 2017–20. We show, using targeted metagenomic sequencing of >100 pathogens, including all common respiratory viruses and bacteria, from samples collected from 433 infants, that burden of additional viruses is common (111/433, 26%) but only modestly correlates with RSV disease severity. In contrast, there is strong evidence in both cohorts and across age groups that presence of *Haemophilus* bacteria (194/433, 45%) is associated with higher severity, including much higher rates of hospitalisation (odds ratio 4.25, 95% CI 2.03–9.31). There is no evidence for association between higher severity and other detected bacteria, and no difference in severity between RSV genotypes. Our findings reveal the genomic diversity of additional pathogens during RSV infection in infants, and provide an evidence base for future causal investigations of the impact of co-infection on RSV disease severity.

Human respiratory syncytial virus (RSV) is the leading cause of hospitalisation associated with acute lower respiratory tract infection (LRTI) in infants and young children worldwide. The global RSV epidemic is estimated to cause 33 million LRTIs annually, leading to 3.6 million hospitalisations and around 100,000 RSV-attributable overall deaths in children under 5 years old, with mortalities predominantly in low- and middle-income countries[1].

To date, the standard of care for RSV infection has been supportive management. No safe and effective antivirals are available for RSV treatment. Palivizumab, a monoclonal antibody for short-term RSV prophylaxis, has been used in young children at high risk for severe RSV LRTI, but it requires monthly administrations and its use is limited to higher-income settings due to prohibitive cost[2]. Nirsevimab, another monoclonal antibody with an extended half-life, has recently

A full list of affiliations appears at the end of the paper. *A list of authors and their affiliations appears at the end of the paper.
✉e-mail: gulung.lin.oxford@gmail.com; tanya.golubchik@sydney.edu.au

been authorised for use to prevent severe RSV LRTI with a single injection in infants during their first RSV season in the European Union, Great Britain, Canada, and the US. It has started to be rolled out to all neonates, infants, and at-risk children <2 years of age[3].

The ReSVinet scale has been developed as a global clinical severity scale with the aim of objectively evaluating infants with respiratory infections, specifically acute bronchiolitis[4]. This scale has been validated for construct validity, inter-rater reliability, and usability, even when used by non-healthcare professionals (e.g., parents) or when using information from medical records. The scale combines seven clinical variables, including feeding intolerance, medical intervention, respiratory difficulty, respiratory frequency, apnoea, general condition, and fever. The scores range from 0 to 20, where higher ReSVinet scores represent more severe disease.

Infants normally harbour commensal microorganisms in the upper respiratory tract regardless of respiratory symptoms, including bacteria with pathogenic potential such as *Streptococcus pneumoniae* and *Haemophilus influenzae*[5], and viruses such as enteroviruses and coronaviruses[6–9]. Simultaneous detection of multiple viruses is frequent in both healthy children[9] and children with respiratory infections[9–15], and virus-virus interactions can be either synergistic or antagonistic. A recent study showed that co-infection with influenza A virus and RSV can lead to the formation of hybrid viral particles in vitro, which evade neutralising antibodies and broaden receptor tropism[16]. In contrast, rhinovirus has been shown to interfere with influenza virus, reducing the chance of co-detection and co-circulation of both viruses at the individual and population levels, respectively[17]. Throughout the first year of life, there are constant changes in the respiratory microbiome with increasing biodiversity[6,18]. Colonisation with *Moraxella*, *Haemophilus*, or *Streptococcus* spp. and pneumococcal disease have been shown to be associated with viral respiratory infections[6,19,20].

Molecular testing and typing of respiratory viruses and common bacterial pathogens have been increasingly used in the clinical setting. However, despite the high prevalence of colonisation and co-infection, the associations between other potential pathogens and the presentation of RSV infection (e.g., disease severity) remain to be understood.

In this study we applied high-throughput RNA-based targeted metagenomic sequencing to simultaneously recover over 100 bacterial and viral pathogens from nasopharyngeal swabs prospectively collected from RSV-infected infants enrolled in two differently designed multicentre studies in Europe. We sought to explore and compare the associations (but not causality) between RSV disease severity and the presence of additional respiratory pathogens in these two infant cohorts. A better understanding of microbial factors associated with RSV disease severity will help direct therapeutic and preventive measures and improve the management and outcome of infants with RSV infection.

## Results
### Study and sample populations
We examined correlates of RSV disease severity in two separately recruited cohorts (from the longitudinal birth cohort study and the infant cross-sectional study), representing two distinct study populations[21,22]. This design enabled us to look for correlates that were robustly present in both studies, reducing the impact of study-specific confounders. The longitudinal birth cohort comprised healthy term infants followed up from birth and sampled whenever even minor respiratory symptoms were present during the RSV seasons before their first birthday. This cohort thus included all prospectively identified infections, resulting in an enrichment for mild disease presentation. In contrast, infants presenting with symptoms and testing positive for RSV were recruited into the infant cross-sectional study, which thus captured a higher-severity cohort.

A total of 440 RSV-positive infants under 1 year of age were enrolled in the two studies, conducted in Spain, the UK, and the

Netherlands during 2017–20 (Supplementary Table 1). Seven infants were excluded from further analysis on the basis of sample quality (discussed below). At least one nasopharyngeal swab was collected from each infant upon testing positive for RSV, with 125 hospitalised infants enrolled in the infant cross-sectional study having daily swabs collected (mean ± SD number of swabs, 4.2 ± 2.5) until hospital discharge.

The demographic and clinical characteristics of the remaining 433 infants (33% from the longitudinal birth cohort study and 67% from the infant cross-sectional study) are shown in Table 1. Due to the differences in study design, infants in the infant cross-sectional study were generally younger and had more severe disease than those in the longitudinal birth cohort study. As expected, younger infants tended to have a higher ReSVinet score and were more likely to require hospitalisation, intensive care, respiratory support, and mechanical ventilation than older infants, whereas a greater proportion of older infants had a fever (<3 months vs. 3 to <6 months vs. 6 to <12 months) (Supplementary Table 2).

### Multi-pathogen sequencing
A total of 839 nasopharyngeal swabs from the 440 RSV-positive infants were sequenced using the *Castanet* custom multi-pathogen enrichment panel (see Methods), with an estimated limit of detection of ~100 copies/mL, as previously reported[23]. One infant's sample yielded zero total reads, consistent with failed library preparation, and no RSV reads were recovered from six infants (seven samples). Among the six infants' seven samples with no RSV reads, five were RSV-negative on re-testing by reverse transcription quantitative PCR (RT-qPCR), one had detectable RSV at 18,463 copies/mL on re-testing by RT-qPCR, and one had no viral load data available. These seven infants and eight samples were excluded from further analysis, which was conducted on the remaining 433 infants and their 831 samples.

Controls are crucial for sequencing experiments, particularly when targeting multiple pathogens. To detect contamination, we included two types of negative controls: nine extraction controls (containing transport media only) and 11 RSV-negative swabs (from patients with respiratory symptoms who tested negative for RSV in this study). All 20 negative controls were interspersed among RSV-positive samples and underwent the entire experimental protocol from extraction to sequencing. None of the extraction controls had reads mapped to reference genomes of viruses and bacteria of interest. The only exception was *Burkholderia multivorans*, detected in three extraction controls—presumably a result of kit contaminants (i.e., kitome), commonly observed in large-scale sequencing studies. This organism was not targeted by *Castanet* and was not included in subsequent analyses. Among the 11 RSV-negative controls, all had at least one viral or bacterial pathogen, as expected for swabs from patients with respiratory symptoms. The most common viruses were rhinovirus ($N = 6$) and human herpesvirus 6 (HHV-6) ($N = 2$), followed by human adenovirus (HAdV), human coronavirus (HCoV) HKU1, influenza A virus, and human parainfluenza virus (one each). The most common bacteria were *Moraxella* spp. ($N = 13$, including *M. catarrhalis*, *M. nonliquefaciens*, and *M. lincolnii*) and *Haemophilus influenzae* ($N = 3$), followed by *Dolosigranulum pigrum*, *Streptococcus mitis*, and *Streptococcus pneumoniae* (one each).

### RSV genotype and viral burden
We previously showed that the number of uniquely mapped RSV reads from our protocol is highly correlated with RSV viral load[24] (two-tailed Pearson correlation, $R^2 = 0.78$, $P = 5 \times 10^{-197}$; Supplementary Fig. 1). Thus, we used the number of unique RSV reads as a surrogate for viral load for the analyses that follow. Of the 831 samples included, 597 (72%) samples had at least 70% of the RSV genome reconstructed with a median of 11,851 unique RSV reads (range, 285 to 810,113), corresponding to a median viral load of

**Table 1 | Characteristics of the 433 RSV-positive infants[a]**

| | Longitudinal birth cohort study (N = 143) | Infant cross-sectional study (N = 290) | P-value | Total (N = 433) |
|---|---|---|---|---|
| Demographic features[b] | | | | |
| Age[c] | | | | |
| Median (IQR) — month | 5.7 (3.6–8.9) | 3.0 (1.5–6.4) | $2.5 \times 10^{-10}$ | 4.1 (1.9–7.5) |
| Distribution | | | $8.3 \times 10^{-10}$ | |
| <3 months | 25/142 (18) | 144/289 (50) | | 169/431 (39) |
| 3 to <6 months | 50/142 (35) | 67/289 (23) | | 117/431 (27) |
| 6 to <12 months | 67/142 (47) | 78/289 (27) | | 145/431 (34) |
| Gestational age | | | | |
| Median (IQR) — week | 39.9 (39.0–40.9) | 39.4 (38.0–40.3) | $9.8 \times 10^{-5}$ | 39.6 (38.6–40.4) |
| Distribution | | | $3.3 \times 10^{-4}$ | |
| <32 weeks | 0/139 (0) | 8/288 (3) | | 8/427 (2) |
| 32 to <37 weeks | 0/139 (0) | 17/288 (6) | | 17/427 (4) |
| ≥37 weeks | 139/139 (100) | 263/288 (91) | | 402/427 (94) |
| Female sex | 67/142 (47) | 125/289 (43) | 0.504 | 192/431 (45) |
| Comorbidity[d] | 0/143 (0) | 37/289 (13) | $1.9 \times 10^{-7}$ | 37/432 (9) |
| Sampling season | | | $7.4 \times 10^{-6}$ | |
| 2017–18 | 13 (9) | 57 (20) | | 70 (16) |
| 2018–19 | 40 (28) | 121 (42) | | 161 (37) |
| 2019–20 | 90 (63) | 112 (39) | | 202 (47) |
| Sampling country | | | $1.9 \times 10^{-13}$ | |
| Spain | 52 (36) | 25 (9) | | 77 (18) |
| United Kingdom | 30 (21) | 137 (47) | | 167 (39) |
| Netherlands | 61 (43) | 128 (44) | | 189 (44) |
| Virological features | | | | |
| RSV-A[e] | 75/142 (53) | 145/282 (51) | 0.207 | 220/424 (52) |
| Peak RSV read count | n = 136 | n = 286 | | n = 422 |
| Mean ± SD — $\log_{10}$ | 4.2 ± 1.0 | 3.9 ± 1.0 | $0.079^f$ | 4.0 ± 1.0 |
| Clinical features[g] | | | | |
| ReSVinet score | | | | |
| Mean ± SD | 4.7 ± 2.6 | 8.9 ± 4.7 | $1.0 \times 10^{-8}$ | 7.6 ± 4.6 |
| Distribution | | | $1.3 \times 10^{-8}$ | |
| 0–7 | 120/136 (88) | 120/284 (42) | | 240/420 (57) |
| 8–13 | 15/136 (11) | 107/284 (38) | | 122/420 (29) |
| 14–20 | 1/136 (1) | 57/284 (20) | | 58/420 (14) |
| Fever | 53/136 (39) | 89/284 (31) | 0.258 | 142/420 (34) |
| Hospitalisation | 7/115 (6) | 212/289 (73) | $5.6 \times 10^{-12}$ | 219/404 (54) |
| PICU admission | 1/115 (1) | 76/289 (26) | 0.003 | 77/404 (19) |
| Respiratory support | 3/112 (3) | 176/260 (68) | $6.0 \times 10^{-10}$ | 179/372 (48) |
| Mechanical ventilation | 0/112 (0) | 66/260 (25) | 0.985 | 66/372 (18) |

[a] Unless otherwise specified, data are shown as number/total number (%) or number (%) if there is no missing value. Percentages may not total 100 due to rounding.

[b] Two-tailed Mann–Whitney U tests were used to compare continuous variables between the two groups; two-sided chi-square tests with Yates' correction or two-sided Fisher's exact tests were used to compare categorical variables between the two groups, whichever is appropriate.

[c] At the time of RSV infection.

[d] Comorbidities included prematurity with or without bronchopulmonary dysplasia, ventricular septal defect, and other congenital abnormalities.

[e] Nine participants with both RSV subgroups A B identified were excluded from this comparison. Two-sided multivariable logistic regression was used to adjust for sampling season.

[f] Two-sided multiple linear regression was used to adjust for the duration between symptom onset and sampling.

[g] Multiple linear regression, ordered logistic regression, or multivariable logistic regression (all two-sided) was used to adjust for covariates, depending on the type of the response (dependent) variable. Covariates included age, gestational age, sex, comorbidity, RSV subgroup, sampling season and country, and peak RSV read count along with the duration between symptom onset and sampling. Models with different combinations of the covariates were tested, and the model with the lowest Akaike information criterion (AIC) was selected.

IQR interquartile range, PICU paediatric intensive care unit, RSV respiratory syncytial virus, SD standard deviation.

$1.2 \times 10^7$ copies/mL (range, 505 to $4.3 \times 10^9$). Peak RSV read counts for the two studies are shown in Table 1.

Similar proportions of the 433 infants had RSV-A and RSV-B (220 vs. 204; the remaining nine could not be genotyped due to evidence of mixed infection or contamination). RSV-B dominated the 2017–18 and 2018–19 RSV seasons, accounting for 71% and 55% of the infections, respectively, whereas RSV-A was the predominant subgroup during the 2019–20 RSV season, accounting for 66% of the infections (Supplementary Table 3). Phylogenetic analyses classified all RSV-A strains into genotype ON1 and all RSV-B strains into genotype BA (Supplementary Fig. 2), as previously described[25]. There was no evidence to support differences in severity between RSV-A and RSV-B infections (Supplementary Table 4).

### RSV and co-detected respiratory viruses

At least one respiratory virus in addition to RSV was recovered in 111 (26%) of the 433 RSV-infected infants, including nine infants with two additional viruses (Fig. 1). Non-RSV viruses were present at substantial viral load: in the quantitative sequencing method we used, 92% of the non-RSV viruses had sufficient reads to enable reconstruction of at least 50% of their targeted genome (covered by at least two reads), consistent with viral loads in a range similar to that of RSV and suggestive of active viral replication/infection (Supplementary Fig. 3). RSV-infected infants with any viral co-detection tended to be older than those without [median (interquartile range or IQR), 4.6 (2.4–8.0) vs. 3.8 (1.7–7.0) months; two-tailed Mann–Whitney U test, P = 0.028] (Supplementary Table 5), primarily driven by infants with HCoV co-detection (Supplementary Fig. 4). There was no correlation between the viral load of non-RSV viruses and RSV viral load or genotype, and likewise no correlation between viral load of non-RSV viruses and infant age, sex, or clinical outcome.

Rhinovirus was the virus most frequently found alongside RSV (68/433 infants, 16%), followed by seasonal HCoVs (3%) and HAdVs (2%) (Supplementary Table 6). We did not find any SARS-CoV-2 in 76 swabs collected from 35 infants in January and February 2020. VP1 genotyping of the rhinovirus detected in the 68 infants showed that rhinovirus A, B, and C accounted for 43% (29/68), 26% (18/68), and 31% (21/68), respectively (Supplementary Fig. 5). HHV-6 was detected in seven (2%) infants, including two HHV-6A and five HHV-6B (Supplementary Fig. 6). Phylogenetic segregation of HHV-6A was strongly suggestive of chromosomally integrated HHV-6A in our samples[26].

### Correlates of RSV disease severity

To understand the relationship between co-detected viruses and RSV disease severity, we defined severity in terms of the ReSVinet score, reported for both study cohorts. Individual clinical outcome variables were also used to define disease severity, including presence of fever, and requirement for hospitalisation, intensive care, respiratory support, and invasive mechanical ventilation. The associations were then examined between detection of additional viruses and either ReSVinet score as a composite measure, or the individual clinical variables separately. We used linear regression, logistic regression, and proportional odds ordered logistic regression models (all two-sided), with adjustment for potential confounders, including age, gestational age, presence of comorbidities, RSV viral load, and study cohort (see Methods for the full list of covariates).

There was evidence that infants with viral co-detection were more likely to require intensive care and mechanical ventilation than those without, after adjusting for potential confounders (Fig. 2 and Supplementary Table 5). Among the confounders, gestational age was a significant covariate in both models (P = 0.002 for intensive care and $7.7 \times 10^{-5}$ for mechanical ventilation), whereas comorbidity was not independently predictive of either outcome. Notably, prematurity accounted for the majority of comorbidities in our study population (24/35, 69%).

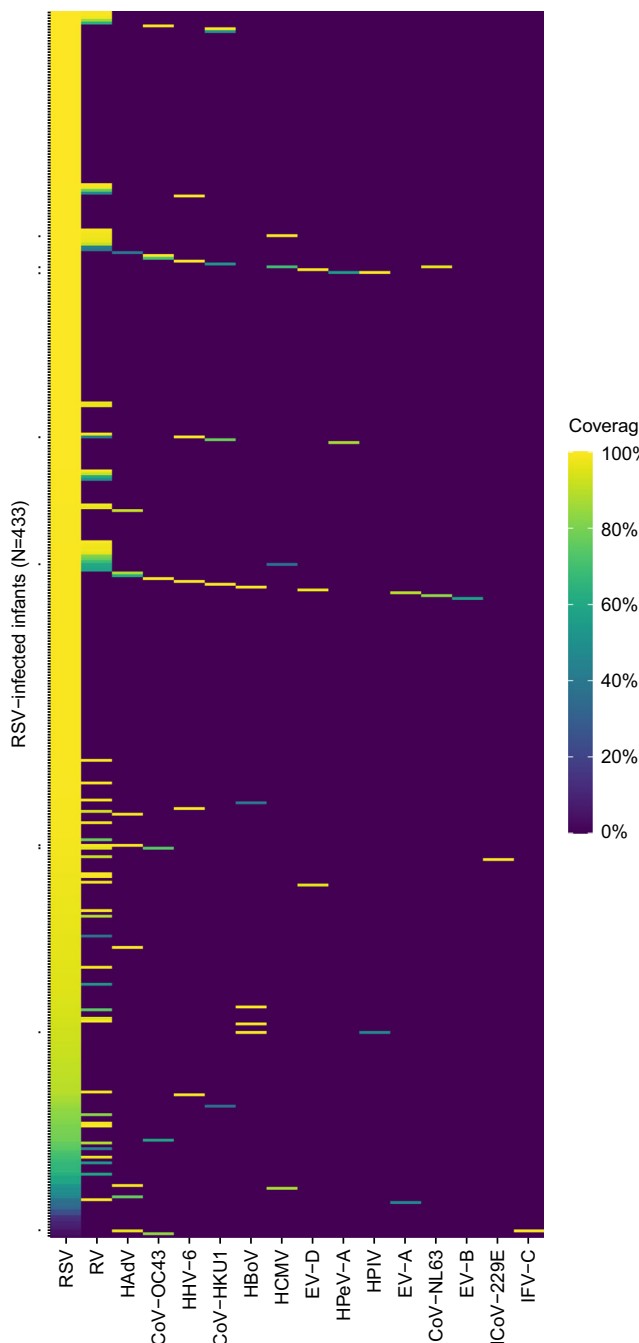

**Fig. 1 | Genome coverage of all detected viruses in RSV-infected infants.** Each row represents an individual infant. Dots on the Y axis denote the nine infants who had two other viruses identified in addition to RSV. Genome coverage is illustrated by the coloured gradient, defined as the proportion of the targeted genome covered by at least two reads. Viruses are ordered in decreasing prevalence from left to right. Source data are provided as a Source Data file. EV enterovirus, HAdV human adenovirus, HBoV human bocavirus, HCMV human cytomegalovirus, HCoV human coronavirus, HPeV-A human parechovirus A, HPIV human parainfluenza virus, IFV-C influenza C virus, RSV respiratory syncytial virus, RV rhinovirus.

Analysing each non-RSV virus family separately (i.e., enterovirus, HCoV, HAdV, and HHV-6; Supplementary Tables 7–10) showed that infants with HCoV co-detection tended to be older than those without [median (IQR), 8.2 (4.2–10.0) vs. 4.0 (1.8–7.3) months] (Supplementary Table 8; Supplementary Fig. 4). A higher proportion of infants with HHV-6 co-detection required intensive care than those without (71% [5/7] vs. 18% [72/397]; odds ratio, 9.4; 95% CI, 1.5–78.9; Cohen's

$f^2 = 0.023$; Supplementary Table 10), but the limited sample numbers of HHV-6 preclude a robust comparison (not significant after adjusting for multiple comparisons).

We found no evidence of genotype-to-genotype association between RSV and the co-detected viruses. Figure 3 shows the distribution of the non-RSV viruses on the RSV phylogenies. The $D$ statistic was used to assess the phylogenetic signal for the non-RSV viruses on RSV phylogenies based on a permutation testing framework (see Methods for interpretation of $D$ values)[27]. There was no significant phylogenetic signal for the presence of these viruses on the RSV phylogenies with $D$ values ranging from 0.71 to 1.30, indicating no specific RSV clade was linked with the presence of other viruses (Supplementary Table 11).

### RSV and co-detected bacterial pathogens

Bacterial species with known respiratory associations were detected in the vast majority of infants (406/433, 94%). The 27 infants without any detected targeted bacteria appeared to have low-quality swabs rather than evidence of recent antibiotic use. Among the 27 infants, 21 had available information on antibiotic usage, out of which 10 did not receive any antibiotics during the RSV infection, 10 had received antibiotics before sample collection, and one received antibiotics without information on the timing of antibiotic dosing. These 27 infants had a mean ± SD peak RSV read count of $3.7 \pm 1.2$ $\log_{10}$, falling in the lower quartile of the study population (Supplementary Fig. 3), suggesting these may have been poorer quality swabs.

A median of three targeted bacterial species (IQR 2–4, range 0–14) were found in each infant. The most frequently found genera were *Moraxella* (76% of all 433 RSV-infected infants), *Streptococcus* (68%), and *Haemophilus* (45%) (Supplementary Table 12, Supplementary Figs. 7 and 8, Supplementary Data 1). These three genera together were found in 392/433 infants (91%). Among each of these genera, *M. catarrhalis*, *S. pneumoniae*, and *H. influenzae* were the most commonly identified species, and had read numbers consistent with substantial bacterial load (Supplementary Fig. 3). Infants with any of these three bacterial genera were older than those without [median (IQR), 4.6 (1.9–7.7) vs. 2.0 (1.1–3.4) months; two-tailed Mann–Whitney U test, $P = 0.002$] (Supplementary Fig. 9).

Information on breastfeeding was available in 12% (51/433) of the infants. No significant differences in co-detected bacteria or clinical outcomes were found between breastfed infants (either exclusively or in combination with formula milk) and exclusively formula-fed infants, based on their feeding status within the 4 weeks prior to the RSV infection (Supplementary Table 13, Supplementary Fig. 10).

Co-detected bacteria in the infants were highly genetically diverse. Five of the recovered bacterial species were further classified down to the ribosomal sequence type (rST) level in some samples: *M. catarrhalis*, *S. pneumoniae*, *H. influenzae*, *Escherichia coli*, and *Neisseria meningitidis*. Most rSTs were unique in the dataset, consistent with extensive bacterial genetic diversity at the population level. Only a few isolates shared the same rST—*M. catarrhalis* rST 105842 was found in two infants; each of the *S. pneumoniae* rSTs 644, 12187, 13906, and 104663 were found in two infants; and each of the *H. influenzae* rSTs 24162, 49634, 89110, and 133371 were also found in two infants (Supplementary Table 14).

### Positive association between *Haemophilus* and RSV severity

Similarly to the analysis of viral co-detection, we investigated the correlation between disease severity and the presence of *Moraxella*, *Haemophilus*, and *Streptococcus*, after adjusting for confounders as above. Compared with infants without any *Haemophilus* spp., infants with *Haemophilus* spp. were more likely to have a higher ReSVinet score and fever and require hospitalisation, respiratory support, and

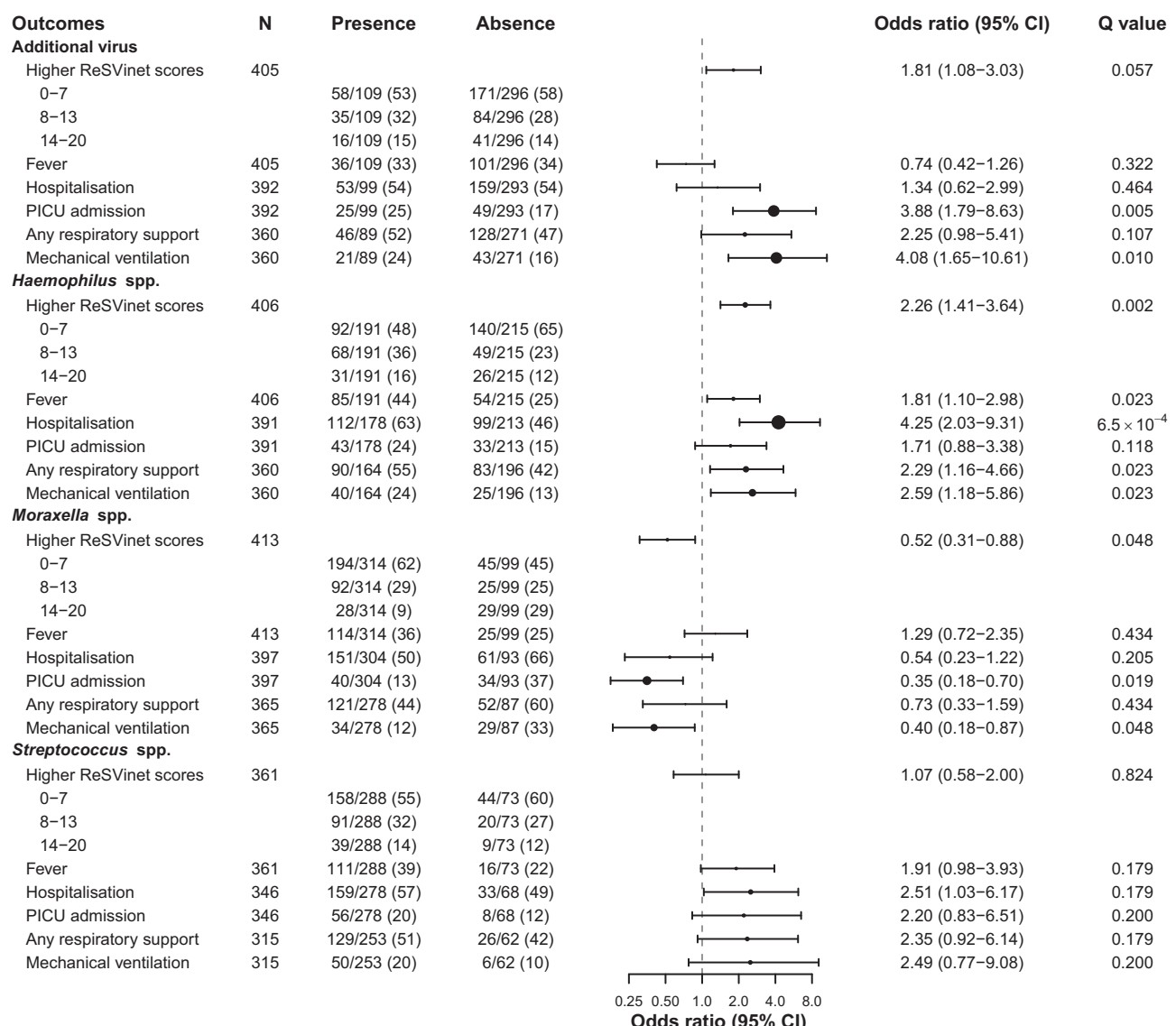

| Outcomes | N | Presence | Absence | Odds ratio (95% CI) | Q value |
|---|---|---|---|---|---|
| **Additional virus** | | | | | |
| Higher ReSVinet scores | 405 | | | 1.81 (1.08–3.03) | 0.057 |
| 0–7 | | 58/109 (53) | 171/296 (58) | | |
| 8–13 | | 35/109 (32) | 84/296 (28) | | |
| 14–20 | | 16/109 (15) | 41/296 (14) | | |
| Fever | 405 | 36/109 (33) | 101/296 (34) | 0.74 (0.42–1.26) | 0.322 |
| Hospitalisation | 392 | 53/99 (54) | 159/293 (54) | 1.34 (0.62–2.99) | 0.464 |
| PICU admission | 392 | 25/99 (25) | 49/293 (17) | 3.88 (1.79–8.63) | 0.005 |
| Any respiratory support | 360 | 46/89 (52) | 128/271 (47) | 2.25 (0.98–5.41) | 0.107 |
| Mechanical ventilation | 360 | 21/89 (24) | 43/271 (16) | 4.08 (1.65–10.61) | 0.010 |
| ***Haemophilus* spp.** | | | | | |
| Higher ReSVinet scores | 406 | | | 2.26 (1.41–3.64) | 0.002 |
| 0–7 | | 92/191 (48) | 140/215 (65) | | |
| 8–13 | | 68/191 (36) | 49/215 (23) | | |
| 14–20 | | 31/191 (16) | 26/215 (12) | | |
| Fever | 406 | 85/191 (44) | 54/215 (25) | 1.81 (1.10–2.98) | 0.023 |
| Hospitalisation | 391 | 112/178 (63) | 99/213 (46) | 4.25 (2.03–9.31) | $6.5 \times 10^{-4}$ |
| PICU admission | 391 | 43/178 (24) | 33/213 (15) | 1.71 (0.88–3.38) | 0.118 |
| Any respiratory support | 360 | 90/164 (55) | 83/196 (42) | 2.29 (1.16–4.66) | 0.023 |
| Mechanical ventilation | 360 | 40/164 (24) | 25/196 (13) | 2.59 (1.18–5.86) | 0.023 |
| ***Moraxella* spp.** | | | | | |
| Higher ReSVinet scores | 413 | | | 0.52 (0.31–0.88) | 0.048 |
| 0–7 | | 194/314 (62) | 45/99 (45) | | |
| 8–13 | | 92/314 (29) | 25/99 (25) | | |
| 14–20 | | 28/314 (9) | 29/99 (29) | | |
| Fever | 413 | 114/314 (36) | 25/99 (25) | 1.29 (0.72–2.35) | 0.434 |
| Hospitalisation | 397 | 151/304 (50) | 61/93 (66) | 0.54 (0.23–1.22) | 0.205 |
| PICU admission | 397 | 40/304 (13) | 34/93 (37) | 0.35 (0.18–0.70) | 0.019 |
| Any respiratory support | 365 | 121/278 (44) | 52/87 (60) | 0.73 (0.33–1.59) | 0.434 |
| Mechanical ventilation | 365 | 34/278 (12) | 29/87 (33) | 0.40 (0.18–0.87) | 0.048 |
| ***Streptococcus* spp.** | | | | | |
| Higher ReSVinet scores | 361 | | | 1.07 (0.58–2.00) | 0.824 |
| 0–7 | | 158/288 (55) | 44/73 (60) | | |
| 8–13 | | 91/288 (32) | 20/73 (27) | | |
| 14–20 | | 39/288 (14) | 9/73 (12) | | |
| Fever | 361 | 111/288 (39) | 16/73 (22) | 1.91 (0.98–3.93) | 0.179 |
| Hospitalisation | 346 | 159/278 (57) | 33/68 (49) | 2.51 (1.03–6.17) | 0.179 |
| PICU admission | 346 | 56/278 (20) | 8/68 (12) | 2.20 (0.83–6.51) | 0.200 |
| Any respiratory support | 315 | 129/253 (51) | 26/62 (42) | 2.35 (0.92–6.14) | 0.179 |
| Mechanical ventilation | 315 | 50/253 (20) | 6/62 (10) | 2.49 (0.77–9.08) | 0.200 |

0.25 0.50 1.0 2.0 4.0 8.0
**Odds ratio (95% CI)**

**Fig. 2 | Clinical outcomes by presence and absence of pathogen groups.** Odds ratios were adjusted for covariates including age, gestational age, sex, comorbidity, sampling season and country, study cohort, RSV subgroup, and peak RSV read count along with the duration between symptom onset and sampling using two-sided ordered logistic regression or two-sided multivariable logistic regression. $Q$ values were adjusted for the false discovery rate using the Benjamini–Hochberg method; a value of less than 0.05 was considered to indicate statistical significance. Data in presence and absence are shown as no./total no. (%). Black dots represent the odds ratio, sized in proportion to Cohen's $f^2$, ranging from $9.5 \times 10^{-5}$ (ReSVinet scores for *Streptococcus* spp.) to 0.066 (hospitalisation for *Haemophilus* spp.). Source data are provided as a Source Data file. CI confidence interval, PICU paediatric intensive care unit.

mechanical ventilation (Figs. 2 and 4a, Supplementary Table 15). Although the effect size of *Haemophilus* presence on most of these clinical outcomes was small (Cohen's $f^2$ ranging from 0.010 to 0.066) (Supplementary Table 15), the presence of *Haemophilus* spp. significantly correlated with a 2.5- and 1.9-point increase in ReSVinet scores (adjusted for only age and adjusted for all confounders, respectively) across all age groups combining both study cohorts (second panel in Fig. 4a).

Examining the two study cohorts separately (bottom two panels in Fig. 4a), the presence of *Haemophilus* spp. was also significantly associated with higher ReSVinet scores across age groups despite different patterns of correlation between age and ReSVinet score within each cohort. As expected on the basis of the difference in study design, ReSVinet scores negatively correlated with age in the infants enrolled in the infant cross-sectional study (two-tailed Pearson correlation, $P = 3.3 \times 10^{-9}$) but not in those in the longitudinal birth cohort study ($P = 0.44$) (bottom two panels in Fig. 4a). The association with

*Haemophilus* was robust to the difference in study population between the cohorts, supporting the separate effect of *Haemophilus* to known effects of age.

We hypothesised that antibiotic administration may have been higher among infants with *Haemophilus* spp., potentially indicating perceived severity of illness or secondary bacterial infection. Information on antibiotic use was available for 361 RSV-infected infants, of whom 91 (25%) received antibiotics during their RSV episode. Only 30 of the 91 infants had documented bacterial isolates clinically (Supplementary Table 16). Consistent with our hypothesis, a higher proportion of infants with *Haemophilus* spp. received antibiotics than those without (33% vs. 20%; two-sided chi-square test, $P = 0.010$). Among infants with *Haemophilus* co-detection, those receiving antibiotics tended to have more severe RSV disease than those without antibiotic treatment, supported by all tested clinical outcome measures except fever (Supplementary Table 17). The positive association between disease severity and *Haemophilus* co-detection described

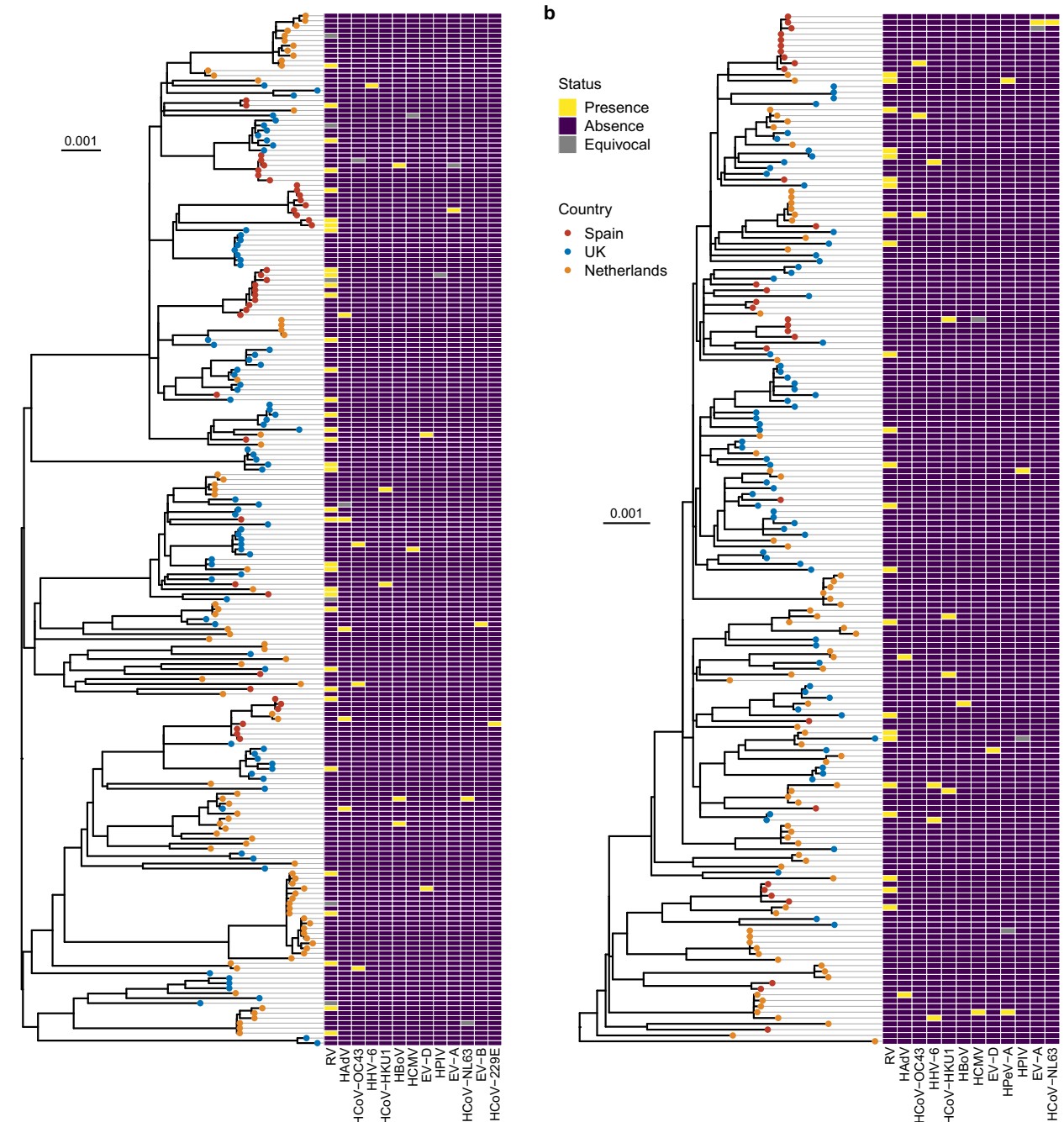

**Fig. 3 | Distribution of co-detected viruses on the RSV maximum-likelihood phylogenies. a** RSV-A phylogeny was reconstructed from 207 samples. **b** RSV-B phylogeny was reconstructed from 177 samples. At least 70% of the coding sequences were recovered from these samples. For infants with multiple samples collected, only the sample with the highest coverage was included. The trees were midpoint rooted. The sampling country of each strain is illustrated by tip colour. The scale bars represent the number of nucleotide substitutions per site. Human parechovirus A (HPeV-A) was not co-detected with any of the RSV-A strains; enterovirus B (EV-B) and human coronavirus 229E (HCoV-229E) were not co-detected with any of the RSV-B strains. Source data are provided as Source Data files. HAdV human adenovirus, HBoV human bocavirus, HCMV human cytomegalovirus, HHV-6 human herpesvirus 6, HPIV human parainfluenza virus, RSV respiratory syncytial virus, RV rhinovirus.

above remained when including antibiotic treatment as one of the confounders in the analysis.

### Negative association between *Moraxella* and RSV severity

In contrast to the observation of higher ReSVinet scores with *Haemophilus* spp., infants with *Moraxella* spp. tended to have lower ReSVinet scores (especially among the overall study population under 6 months of age and among those in the infant cross-sectional study; Fig. 4b) and were also less likely to require intensive care and mechanical ventilation than those without (Fig. 2). Infants with and without *Streptococcus* spp., or *S. pneumoniae* in particular, had similar clinical outcomes (Fig. 2), despite a modest increase in the ReSVinet score among the overall study population with *Streptococcus* spp. after adjusting for age (Fig. 4c).

In terms of antibiotic usage, a lower proportion of infants with *Moraxella* spp. received antibiotics than those without (19% vs. 43%; $P = 2.2 \times 10^{-5}$). The rates of receiving antibiotics were similar in infants with and without *Streptococcus* spp. (22% vs. 28%; $P = 0.402$), but fewer

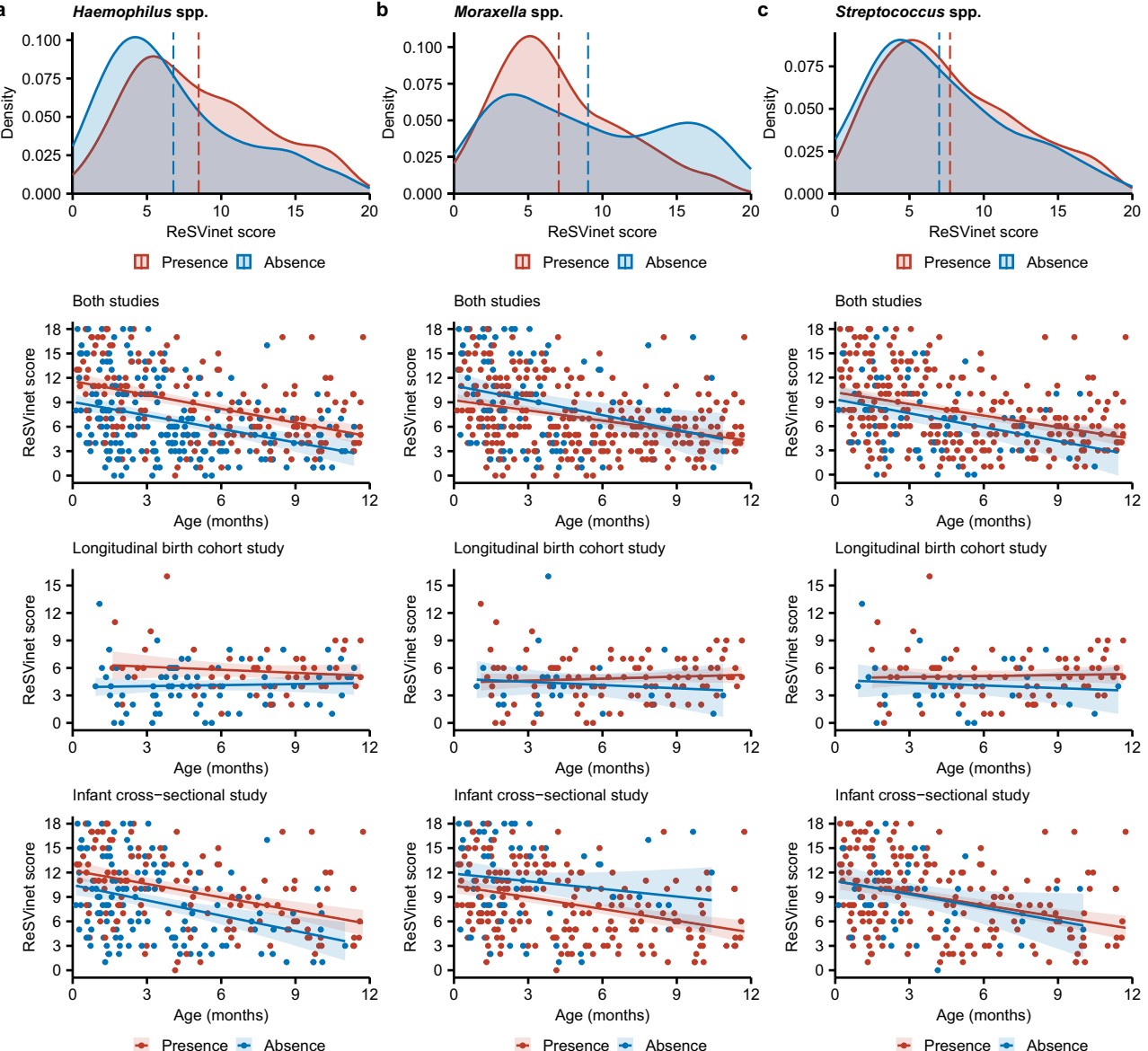

**Fig. 4 | Density plots and correlations between ReSVinet score and age, stratified by presence of three most common bacterial genera.** On the density plots, dashed lines represent the mean ReSVinet score. On the scatter plots, each dot represents an individual infant, and the lines are the simple linear regression lines with the shaded area representing the 95% confidence interval. Two-sided one-way analysis of covariance was performed to examine the differences between the regression lines. No adjustments for multiple comparisons were made. Results of the correlations from the longitudinal birth cohort study and the infant cross-sectional study are shown both in aggregate and separately. **a** *Haemophilus* species. Presence of *Haemophilus* significantly correlated with a 2.5-point increase in ReSVinet scores among all study population [$N = 405$, $F(1, 402) = 34.2$, $P = 1.0 \times 10^{-8}$], a 1.6-point increase in the longitudinal birth cohort study [$N = 133$, $F(1, 130) = 12.3$, $P = 6.3 \times 10^{-4}$], and a 2.1-point increase in the infant cross-sectional study [$N = 272$, $F(1, 269) = 14.5$, $P = 1.7 \times 10^{-4}$]. **b** *Moraxella* species. Presence of *Moraxella* significantly correlated with a 1.1-point decrease in ReSVinet scores among all study population [$N = 412$, $F(1, 409) = 4.7$, $P = 0.031$] and a 2.0-point decrease in the infant cross-sectional study [$N = 277$, $F(1, 274) = 9.9$, $P = 1.8 \times 10^{-3}$]. However, the difference was not significant in the longitudinal birth cohort study [$N = 135$, $F(1, 132) = 0.9$, $P = 0.348$]. **c** *Streptococcus* species. Presence of *Streptococcus* significantly correlated with a 1.3-point increase in ReSVinet scores among all study population [$N = 360$, $F(1, 357) = 5.1$, $P = 0.025$]. However, the difference was not significant in either the longitudinal birth cohort study [$N = 114$, $F(1, 111) = 2.8$, $P = 0.099$] or the infant cross-sectional study [$N = 246$, $F(1, 243) = 0.04$, $P = 0.845$]. Note, unlike the analyses in Supplementary Table 15 and for the other clinical outcomes, these analyses did not adjust for other potential confounders except for age. Source data are provided as a Source Data file.

infants with *S. pneumoniae* received antibiotics than those without (19% vs. 30%; $P = 0.021$).

Supplementary Fig. 11 shows the distribution of commonly co-detected bacterial species (namely *M. catarrhalis*, *S. pneumoniae*, *H. influenzae*, and *Staphylococcus aureus*) on the RSV phylogenies. *S. pneumoniae* was non-randomly distributed across the RSV-B phylogeny ($P = 0.022$) with a $D$ value of 0.77, but the distribution was significantly different from the expectations under Brownian motion ($P < 0.001$), indicating a low phylogenetic signal (Supplementary Table 11). There was no significant phylogenetic signal between the RSV phylogenies and the presence of other commonly co-detected bacteria, with $D$ values ranging from 0.76 to 1.01.

## Discussion

Using RNA-based quantitative sequencing together with a comprehensive and sensitive targeted metagenomic approach capturing over 100 bacterial and viral pathogens, we assessed a broad range of respiratory pathogens in infants with RSV, with the aim of examining

disease severity correlates. We found strong genomic evidence for presence of at least one additional virus in 26% of the 433 RSV-infected infants, aligning with reported viral co-detection rates of 10–65% in children of all ages with respiratory infections[12,28–34]. This co-detection rate is likely a conservative estimate since our stringent detection criteria would necessarily miss very low viral load co-detection. We additionally assessed the presence of respiratory bacteria with high pathogenic potential and detected *Moraxella*, *Streptococcus*, and *Haemophilus* in 91% of the infants, surpassing culture-based methods[35].

It is generally believed that viral co-infection rates decrease with age—infants and children have a higher rate of viral co-infections than adults[28,31,36,37]. However, our observation at the lower extreme of age, under 12 months, challenges this. Older infants were more likely to have multiple respiratory viruses than younger infants. This may be due to maternal antibodies in the youngest infants and increased social contacts in older infants (e.g., attending childcare settings), exposing them to more respiratory viruses.

Rhinovirus was the most commonly co-detected virus, found in 16% of the RSV-infected infants. The Pneumonia Etiology Research for Child Health (PERCH) study identified rhinovirus as the second most common virus causing severe pneumonia in children after RSV, and the most common virus in healthy children or those without severe pneumonia[9]. The rate of rhinovirus detection in our study was consistent with that in a previous study of RSV co-infections in the US, which reported evidence of viral interference between RSV and rhinovirus[38]. Apart from rhinovirus, no other single virus was detected in >5% of the infants.

Co-detected viruses may indicate either residual nucleic acid from a previous infection or active replication of an additional virus (newly acquired or reactivated). This additional virus could be either a non-pathogenic bystander or a true disease-causing agent. Although not definitive, unique read count as a surrogate for viral load may help differentiate between these possibilities. Co-detected viruses with low viral load may come from prolonged shedding from a previous infection, whereas those with high viral load are more likely to cause symptoms[39]. In this study, we focused on viruses with substantial viral load based on read numbers, indicating active replication and potential co-infection.

In our study population, additional respiratory viruses alongside RSV correlated with an increased need for intensive care and mechanical ventilation. However, the overall correlation between viral co-detection and severity was small, without any single co-detected virus associated with disease severity. Previous studies have disagreed on the potential role of co-infections in RSV severity. For example, Yoshida et al. reported an increased LRTI risk among 401 RSV-infected children <5 years with co-detection of rhinovirus, human metapneumovirus, or human parainfluenza virus 3[12], and da Silva et al. found RSV and rhinovirus co-detection associated with longer hospital stays and oxygen use than RSV alone among 260 children <3 years[32]. In contrast, Chu et al. found no association between severity and viral co-detection in 106 RSV-infected infants[33]; similarly, a meta-analysis of 26 studies in RSV-infected children <5 years found no difference in severity between individuals with and without viral co-infection, except a higher need for intensive care with co-infection of RSV and human metapneumovirus[40]. These differences are likely driven by variations in study design, populations, outcome measures, and analysis methods. Our study focused exclusively on infants (at the highest risk of RSV infection among paediatric populations), comprised one of the largest cohorts to be analysed to date, and utilised multiple outcome measures to address the heterogeneity of clinical presentation—all of which strengthen the clinical significance of our results. Our observation that infants with existing comorbidities were overrepresented among those with viral co-detections suggests a potential host-specific role for harbouring multiple pathogens, warranting further investigation.

*Haemophilus* presence was significantly associated with severe RSV disease, irrespective of age, study cohort, or RSV genotype. Although interpreting the clinical significance of a 1.9-point increase in ReSVinet scores with *Haemophilus* may be challenging, this association was supported by various clinically relevant outcomes, such as the need for hospitalisation and respiratory support, including mechanical ventilation. This finding aligns with previous studies[6,41,42]. Chu et al., employing qPCR array technology and a global respiratory severity score, observed a correlation between *H. influenzae* abundance and severity but found no association with *S. pneumoniae* in RSV-infected infants[33]. Additionally, another study linked elevated *Haemophilus* levels with early recurrence of respiratory symptoms following LRTI[43]. The increased severity with *H. influenzae* overrepresentation has been associated with elevated CD4+ and CD8+ T-cell signatures[33], enhanced Toll-like receptor signalling[41], mucosal chemokine (C-X-C motif) ligand 8 (CXCL8, also known as interleukin [IL]−8) responses[44], IL-17A signalling[41], and inflammatory responses[45]. CXCL8 and IL-17A signalling contribute to macrophage and neutrophil activation and recruitment[41], inducing bronchoalveolar neutrophil infiltration[46].

In line with Chu et al.[33], we found no association between *Streptococcus* presence (or *S. pneumoniae* specifically) and the tested clinical outcomes. Other studies, however, linked *Streptococcus*-dominated microbiota with an increased likelihood of hospitalisation in 106 infants with RSV infection[41] and a greater likelihood of developing LRTI in 184 infants with acute respiratory infection[6], the latter finding being independent of RSV. Additionally, *Streptococcus* abundance has been associated with respiratory infection when comparing microbiome profiles in healthy infants[6,47]. Resolving the role of *S. pneumoniae* conditional on RSV infection requires future work.

Despite being the most frequently found bacteria, *Moraxella* spp. were associated with less severe RSV infection that did not require intensive care or mechanical ventilation in our study, suggesting a potential protective effect or their role as a non-pathogenic bystander. Previous studies also showed a higher frequency of *Moraxella* in outpatients than hospitalised children[41] and in children without severe pneumonia (including healthy children) than those with[9], along with a significant association with fever in infants with RSV LRTI[6]. Although fever is a parameter in the ReSVinet score, we demonstrated elsewhere that it did not correlate with other clinical outcome measures in a subset of these RSV-infected infants[23]. In this study, older infants were more likely to have a fever but generally presented with a milder RSV infection than younger infants. Further studies are required to evaluate the underlying immune mechanisms contributing to the potential protective effect of *Moraxella*.

Our study has several limitations. Regional and age-specific differences in respiratory flora may limit generalisability of our results to infants in low- and middle-income countries or different age groups (e.g., older adults). In addition, the observed profile of co-detected upper respiratory pathogens may not directly reflect the lower airway profile; however, some studies suggest a similarity between the upper and lower airway microbiota[48,49]. Although our method covers a comprehensive list of viruses, co-detection does not imply capacity to initiate infection in the absence of RSV. For instance, we cannot determine whether co-detected HHV-6 was a primary respiratory pathogen or reactivation of a latent infection. More broadly, associations between disease severity and pathogen co-detection should be interpreted with caution and do not indicate causation. Such indication would only come from a therapeutic trial. Antibiotic usage is common in infants with severe RSV infection, affecting the respiratory microbiome, but information on the number of antibiotic doses administered or completion of a full antibiotic course was lacking, preventing further investigation into its impact. Additionally, the ribosomal multilocus sequence typing (rMLST) scheme classified bacteria to the species or rST level, but associating rST with serotype (e.g., for *S. pneumoniae* and *H. influenzae*) remains challenging. Lastly,

although multiple commensal and pathogenic viruses and bacteria were recovered, we focused on targeted organisms and did not attempt to characterise the entire microbial community. Microbiome analyses using samples collected from these infants are ongoing.

This study deepens our understanding of associations between RSV disease severity and co-detected respiratory pathogens, highlighting the potential of multi-pathogen sequencing to illuminate complex polymicrobial infections. Targeted metagenomic panels such as *Castanet* enable simultaneous detection of combinations of specific bacteria and viruses, potentially critical in RSV and other infections. Our results may assist in identifying patients at risk for severe RSV infection, directing therapeutic and prophylactic development, and improving the management and outcomes of individuals infected with RSV. Future studies on transcriptomic and immunological profiles of RSV infection should consider the presence of other bacterial and viral pathogens.

## Methods

### Study design and clinical data collection

Nasopharyngeal swabs were prospectively collected from infants under 1 year of age with primary RSV infection from the community and hospitals in Spain, the UK, and the Netherlands during the 2017–20 RSV seasons. These infants were enrolled in two clinical studies of the Respiratory Syncytial Virus Consortium in Europe (RESCEU) project, a European multicentre project investigating epidemiology, immunology, and virology of RSV infection. The two studies have been previously described in detail[21,22].

In the longitudinal birth cohort study (ClinicalTrials.gov identifier: NCT03627572), healthy term neonates (born at ≥37 weeks' gestation) were enrolled. Key exclusion criteria were a history of major congenital defects (e.g., congenital heart and/or lung disease, genetic, immunological and/or metabolic disorders), acute severe medical conditions (e.g., sepsis or asphyxia), or receipt of immunoglobulin, monoclonal antibodies, or an investigational vaccine or medication against RSV. When participants developed respiratory symptoms during the RSV seasons before their first birthday, a nasopharyngeal swab was taken within 3 days of symptom onset and tested for RSV using point-of-care qualitative molecular testing on the Alere™ i RSV assay (Abbott, Illinois, USA). Regardless of the results, the swabs were immersed in M4RT® transport medium after collection, aliquoted, and frozen at −80 °C until use.

In the infant cross-sectional study (ClinicalTrials.gov identifier: NCT03756766), RSV-infected infants under 1 year of age were enrolled from the community within 4 days of symptom onset and from hospitals within 2 days of admission during the RSV seasons. Infants who were previously healthy and who had pre-existing medical conditions were both eligible for inclusion in this study. Key exclusion criteria were a history of RSV infection, receipt of immunoglobulin or monoclonal antibodies, or exposure to an RSV investigational vaccine or medication. RSV infection was diagnosed using the Alere™ i RSV assay in community settings or by routine antigen or polymerase chain reaction (PCR) tests at the hospital laboratory in hospital settings. A nasopharyngeal swab was collected at the time of enrolment for every participant. For hospitalised participants, daily nasopharyngeal swabs were also collected until hospital discharge. These swabs were also frozen at −80 °C until use.

Demographic and clinical information was gathered in both studies. Clinical outcome variables representing disease severity included the ReSVinet score, presence of fever, and requirement for hospitalisation, intensive care, respiratory support, and invasive mechanical ventilation. All these outcome variables were tested for associations with the presence of co-detected pathogens. The ReSVinet score was calculated by the study staff when the infants were first seen for the acute RSV infection. Fever was defined as at least one episode of a tympanic or rectal temperature of ≥38 °C during the acute infection. Requirement for intensive care was defined as admission to a high

dependency unit or an intensive care unit. Requirement for respiratory support was defined as use of any oxygen delivery device.

The studies were conducted in accordance with the provisions of the Declaration of Helsinki and were approved by the relevant authorities and ethics committees at each site: Hospital Clínico Universitario de Santiago de Compostela, Comité de Ética de la Investigación de Santiago-Lugo (no. 2017/395) in Spain; the University of Oxford, the Health Research Authority (no. 231136), the NHS National Research Ethics Service Oxfordshire Research Ethics Committee A (no. 15/SC/0335) and South Central – Hampshire A (no. 17/SC/0522) in the UK; and the Medical Ethical Committee, University Medical Center Utrecht (no. 17/563) in the Netherlands. The parents or guardians of all participants provided written, informed consent. Reimbursement was only offered where the participant had to travel to the clinic site for a study visit.

### Nucleic acid isolation and targeted metagenomic sequencing

The nasopharyngeal swabs used for sequencing differed from those tested by the initial point-of-care testing or the hospital's antigen and PCR assays for RSV. Nucleic acid isolation and sequencing were performed as described previously[23–25]. The NucliSENS® easyMAG® system (BioMérieux, Marcy-l'Étoile, France) was used for automatic total nucleic acid extraction from 500 μL of each sample, following the manufacturer's instructions. Sequencing libraries were constructed using the SMARTer® Stranded Total RNA-Seq Kit v2 - Pico Input Mammalian (Takara Bio USA, California, USA), following a modified veSEQ-HIV protocol[24,25,50]. A 10-μL aliquot of each library was pooled together, and 750 ng of the pooled library was pulled down with a predesigned SureSelect RNA Target Enrichment multi-pathogen probe set (Agilent, California, USA). This probe set, *Castanet*, consisting of 120-mer oligonucleotides, was designed using the algorithm devised in the veSEQ protocol[51]. It targeted more than 100 potentially pathogenic bacteria and viruses, including both RSV-A and RSV-B (Supplementary Table 18)[52]. The post-capture libraries were amplified with 16 cycles of PCR, and then purified using AMPure XP.

Sequencing was performed on the Illumina MiSeq platform (Illumina, California, US) or the Illumina NovaSeq 6000 system, generating paired-end reads. The MiSeq platform was used for a batch of ≤96 samples, and the NovaSeq 6000 system was used for a batch of 384 samples. Each 96-well sequencing plate also included one RSV-negative sample (collected from participants with acute respiratory symptoms, testing negative by the Alere™ i RSV assay) and one pure M4RT® transport medium (i.e., no template control) as negative controls. These negative controls were processed alongside RSV-positive samples from nucleic acid extraction to library preparation and sequencing; therefore, the no template controls were true extraction controls.

Sequencing was conducted on distinct samples and no samples were sequenced repeatedly. No samples collected from healthy infants were sequenced in this study.

### Viral load measurement

Viral load was determined on the same nasopharyngeal swabs used for sequencing (but different aliquots), using RT-qPCR assays performed at GSK as previously described (protocol proprietary)[53]. The primers of this duplex RT-qPCR assay targeted the N gene for both RSV-A and RSV-B. The limit of detection was 304 and 475 copies/mL for RSV-A and RSV-B, respectively.

### Viral genome assembly and phylogenetic reconstruction

RSV genomes were reconstructed using shiver[54] as previously described[24,25]. Briefly, reads were trimmed to remove adaptors, random primers, and low-quality bases using Trimmomatic (v0.39)[55] (option: Adaptors:2:10:7:1:true LEADING:20 TRAILING:20 SLIDINGWINDOW:4:20 MINLEN:50). Trimmed reads were assembled into contigs using IVA (v1.0.8)[56] and metaSPAdes (v3.14.1)[57], and mapped to genotype-specific RSV references using shiver, with Bowtie 2 (v2.4.1)[58]

as the mapper (option: --very-sensitive-local --maxins 2000 --no-discordant --no-unal). Properly paired reads were retained and duplicates removed using Picard MarkDuplicates (v2.18.14, https://broadinstitute.github.io/picard/). Consensus sequences were generated by shiver, where base calling was supported by a minimum of two unique reads per position.

RSV consensus sequences covering at least 70% of the coding sequences were used to reconstruct phylogenetic trees. For infants with multiple samples collected, only the sample with the highest coverage of the coding sequences was included. To put the study strains in a global context, we also included a subset of contemporaneous global RSV strains downloaded from GenBank on 4th December 2020 with at least 70% coverage of the coding sequences, collected between 2015 and 2019. Genomic sequences were aligned using mafft (v7.490)[59] with the FFT-NS-i method[60]. RAxML (v8.2.12)[61] was used to reconstruct the maximum-likelihood phylogenies with the general time reversible nucleotide substitution model and gamma-distributed rate heterogeneity among sites. The R package ggtree (v2.2.4)[62] was used for tree visualisation. Patristic distances, used to measure phylogenetic distances between sample pairs, were computed using the cophenetic function in the R package stats (v4.0.2). Maximum-likelihood phylogenies of enterovirus and human herpesvirus 6 (HHV-6) were reconstructed using RAxML with the same models as above.

### Respiratory viral genome reconstruction

The *Castanet* probe set[52] is designed to cover the known genetic diversity of respiratory viruses, as well as phylogenetically informative sequences of respiratory-associated bacteria. It includes full genomes of viruses <40 kb in length, and 20 kb of sequence of longer genomes such as human herpesviruses, representing around 10% of the genome from the U23 to U37 genes. Although designed prior to the emergence of severe acute respiratory syndrome coronavirus 2 (SARS-CoV-2), this probe set includes sufficient coronavirus sequences to capture 23 kb of the 29-kb genome of SARS-CoV-2 (Supplementary Fig. 12)[63]. The *Castanet* pipeline (Golubchik, https://github.com/tgolubch/castanet) was used to determine coverage of all targeted viruses[52]. Reads originating from human cells were removed using BBMap (v2020-02-13; Bushnell, https://sourceforge.net/projects/bbmap/). The remaining host-depleted reads were mapped using Bowtie 2 to the references that covered known sequence variability of the targeted viruses. Read depth, genome coverage, and consensus sequences were generated for each virus from binary alignment/map (BAM) files.

Viruses with >35% genome recovery were considered present, and those with <15% genome recovery were considered absent. Viruses fulfilling neither criterion were considered equivocal (i.e., 15–30% genome recovery). Genome recovery was defined as the proportion of the genome covered by a minimum of two reads. The detections were cross-checked using taxonomic classification data generated from Kraken 2 (v0.39)[64]. Specifically, all viruses that were identified as present on the basis of the threshold criteria above had corresponding reads classified as the same virus by Kraken 2, with the exception of enteroviruses. This is due to Kraken 2's very limited enterovirus library, making it less sensitive to this genetically diverse virus[65].

Enterovirus genotyping was performed using the web-based Enterovirus Typing Tool[66] (https://www.rivm.nl/mpf/typingtool/enterovirus/), based on phylogenetic analysis of the VP1 gene. The genotypes of HHV-6 were determined by phylogenetic analysis of the HHV-6 partial genome consensus sequences and reference strains. These references included HHV-6A (strains: U1102, GS, and AJ; accession numbers: NC_001664, KC465951, and KP257584), HHV-6B (strains: Z29 and HST; accession numbers: NC_000898 and AB021506), chromosomally integrated HHV-6A (accession numbers: MG894371 and KY315540), and chromosomally integrated HHV-6B (accession numbers: KY316051 and MG894376).

### Respiratory bacterial reconstruction

The *Castanet* probe set also covered the known variation of 53 *rps* genes of the targeted bacteria. These genes encode the bacterial ribosome protein subunits, and their allele characterisation can be used to classify bacteria into groups at all taxonomic and most typing levels[67]. A database containing catalogued *rps* gene variation has been developed for ribosomal multilocus sequence typing (rMLST)[67].

Reconstruction of the bacterial genetic sequences was based on the rMLST scheme. Contig sequences assembled by IVA and metaSPAdes were matched against all known alleles for each *rps* gene to find exact matches. The PubMLST multi-species isolate database, integrating curated allelic and species information, was then searched to identify the matching species (algorithm available at https://pubmlst.org/species-id)[67]. The PubMLST RESTful application programming interface was applied to efficiently analyse all samples[68,69].

Bacteria supported by at least two exact and unique allelic matches were considered present. A unique allelic match is an allelic pattern that is linked to exactly one taxon at its lowest taxonomic rank or two bacteria with one being an unspecified species of the same genus as the other specified species (e.g., if *Moraxella* sp. and *Moraxella catarrhalis* were both linked to an allelic pattern, *Moraxella catarrhalis* was considered a unique match). Bacteria supported by only one exact and unique match were considered equivocal. The rest were considered absent. A Krona chart was used to visualise the composition of identified bacteria[70].

### Statistical analyses

For infants with multiple samples collected, species that were detected in at least one sample collected while RSV was detectable or within 7 days of enrolment (whichever is longer) were considered present; species that were not detected in any sample collected while RSV was detectable or within 7 days of enrolment (whichever is longer) were considered absent; the rest were considered equivocal. Infants with an equivocal presence of a species were excluded from clinical outcome comparisons.

Continuous variables were summarised using mean, median, standard deviation, and interquartile range. All comparisons of continuous variables between groups were conducted by two-tailed Mann–Whitney U tests (two groups) or Kruskal–Wallis tests (more than two groups); two-sided Dunn's post hoc tests with the Benjamini–Hochberg method were used for multiple-group comparisons. Two-sided chi-square tests with Yates' continuity correction and two-sided Fisher's exact tests were performed for contingency analysis. Two-tailed Pearson correlation analysis was used to evaluate the relationship between two variables.

Phylogenetic signal for a co-detected species on RSV phylogenies was evaluated using the *D* statistic based on a permutation testing framework[27]. An equivocal presence status was considered absence in this analysis. A *D* value of 0 indicates a strong phylogenetic signal under the Brownian motion model of evolution (i.e., conserved trait evolution), and a value of 1 indicates no phylogenetic signal (i.e., a random phylogenetic distribution). A negative *D* value indicates that the binary trait is more conserved than the expectations of the Brownian motion model, whereas a value greater than 1 indicates phylogenetic overdispersion. The *D* statistical tests were performed using the phylo.d function in the R package caper (v1.0.1).

When comparing clinical outcomes between different groups of infants, multiple linear regression (for continuous outcome variables), multivariable logistic regression (for dichotomous outcome variables), and proportional odds ordered logistic regression (for ordered outcome variables) were applied to adjust for covariates using the lm and glm functions in the R packages stats (v4.2.1) and the polr function in the R package MASS (v7.3.60), respectively (all two-sided). When there were more than two groups in the comparisons, likelihood-ratio tests were used to evaluate the effect of the group variable on the goodness of fit of these models. Covariates included age, gestational age, sex,

comorbidity, sampling season and country, study cohort, RSV subgroup, and peak RSV read count along with the duration between symptom onset and sampling. Sex was determined following external examination of body characteristics. Models with different combinations of covariates were tested, and the model with the lowest Akaike information criterion (AIC) was selected. A post hoc adjustment for multiple comparisons with the Benjamini−Hochberg method[71] was applied to determine false discovery rate−corrected $Q$ values in all clinical outcome comparisons. The effect size of the presence of a pathogen on a clinical outcome was evaluated using Cohen's $f^2$. An $f^2$ value between 0.02 and <0.15 represents a small effect size; a value between 0.15 and <0.35, medium; and a value of ≥0.35, large[72].

Two-sided one-way analysis of covariance was performed using the Anova function in the R package car (v3.1.2) to compare ReSVinet scores between presence and absence of the tested bacterial genus across age groups.

All statistical analyses were performed using R (v4.0.2)[73].

### Reporting summary

Further information on research design is available in the Nature Portfolio Reporting Summary linked to this article.

## Data availability

Sequencing data generated in this study have been deposited in the European Nucleotide Archive with the accession code PRJEB34042. The RSV consensus sequences included in the phylogenetic analyses have been deposited in GenBank with the accession numbers shown in Supplementary Table 19. The PubMLST multi-species isolate database, integrating curated allelic and bacterial species information, used to identify bacterial species, is available at https://pubmlst.org/species-id. Source data are provided with this paper, with patients' age shown in 30-day increments and gestational age shown as term/preterm to prevent the identification of individuals.

## Code availability

The R code used for the descriptive statistics and statistical analyses in each table is available at https://doi.org/10.5281/zenodo.10626081 and licensed under a Creative Commons Attribution 4.0 International License[74].

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

## Acknowledgements

This work was supported by the National Institute for Health and Care Research (NIHR) Oxford Biomedical Research Centre, the NIHR Thames Valley and South Midlands Clinical Research Network, the British Research Council, and the Respiratory Syncytial Virus Consortium in Europe (RESCEU) project. RESCEU has received funding from the Innovative Medicines Initiative 2 Joint Undertaking (grant number 116019). This Joint Undertaking receives support from the European Union Horizon 2020 Research and Innovation Program and European Federation of Pharmaceutical Industries and Associations. The views expressed in this article are those of the authors and may not be understood or quoted as being made on behalf of or reflecting the position of the organisations with which the authors are employed/affiliated.

## Author contributions

G.-L.L., T.G., and A.J.P. conceived and designed the work. G.-L.L., S.B.D., M.D.S., J.A., L.B., P.J.M.O., F.M.-T., H.N., A.J.P., and RESCEU Consortium were responsible for the conduct and supervision of the clinical studies. M.A.A. designed the probe set that was used for capture. J.E.B. and K.A.J. designed the algorithm for bacterial identification. M.d.C., D.B., and R.B. designed the sequencing protocol. G.-L.L., A.B., G.M.-C., E.M.-G., and M.d.C. performed the experiments. G.-L.L., T.G., D.O'C., M.A.A., J.E.B., and K.A.J. analysed and interpreted the data. G.-L.L. drafted the manuscript, and all other authors critically revised it. All authors have approved the submitted version and agreed to submit the manuscript. All authors agree to be accountable for all aspects of the work in ensuring that questions related to the accuracy and integrity of any part of the work are appropriately investigated and resolved.

## Competing interests

S.B.D. has been an investigator for clinical trials of vaccines and antimicrobials for pharmaceutical companies, including AstraZeneca, Merck, Pfizer, Valneva, Iliad, Sanofi, and Janssen, and previously sat on RSV advisory boards for Sanofi and Merck. M.D.S. has been an investigator on behalf of the University of Oxford for studies funded or supported by vaccine manufacturers, including Pfizer, GSK, Novavax, MCM vaccines and Janssen. M.D.S. is currently an employee of Moderna Biotech UK, a position he commenced subsequent to the completion of this study. M.A.A. is supported by a Sir Henry Dale Fellowship, jointly funded by the Royal Society and Wellcome Trust (220171/Z/20/Z). J.E.B. and K.A.J. are funded on a Wellcome Trust Biomedical Resources Grant (218205/Z/19/Z, PubMLST: Disseminating and exploiting bacterial diversity data for public health benefit). J.A. was an employee of Janssen Pharmaceutica NV when the work was being conducted. P.J.M.O. has received honoraria from GSK, Pfizer Inc, Sanofi Pasteur, Seqirus, and Janssen for taking part in advisory boards and expert meetings and for acting as a speaker in congresses outside the scope of the submitted work. P.J.M.O. is also a National Institute for Health and Care Research (NIHR) Senior Investigator. F.M.-T. has received honoraria from GSK, Pfizer Inc, Sanofi Pasteur, MSD, Seqirus, and Janssen for taking part in advisory boards and expert meetings and for acting as a speaker in congresses outside the scope of the submitted work. F.M.-T. has also acted as principal investigator in randomised controlled trials of the above-mentioned companies as well as Ablynx, Regeneron, Roche, Abbott, Novavax, and MedImmune, with honoraria paid to his institution. F.M.-T. receives support for his research activities from the Instituto de Salud Carlos III (Proyecto de Investigación en Salud, Acción Estratégica en Salud): Fondo de Investigación Sanitaria (FIS;PI1601569/PI1901090/PI22/00406) del plan nacional de I+D+I and 'fondos FEDER'. H.N. received grants from Innovative Medicine Initiative, NIHR, Bill and Melinda Gates Foundation, WHO, and Pfizer. H.N. also received consultancy or honoraria and speaker fees from Sanofi, Merck, Novavax, ReViral, and GSK (all paid to institution). T.G. is supported by an Investigator Grant (GNT2025445) from the National Health and Medical Research Council, Australia (NHMRC). A.J.P. is chair of the UK Department of Health and Social Care's Joint Committee on Vaccination and Immunisation. A.J.P. has also provided advice to Shionogi on COVID-19 vaccines and his institution receives research funding from NIHR, Bill & Melinda Gates Foundation, Wellcome Trust, Medical Research Council, and AstraZeneca for vaccine research. The remaining authors declare no competing interests.

## Additional information

[1]Oxford Vaccine Group, Department of Paediatrics, University of Oxford, Oxford, UK. [2]NIHR Oxford Biomedical Research Centre, Oxford, UK. [3]Centre for Neonatal and Paediatric Infection, Institute for Infection and Immunity, St George's, University of London, London, UK. [4]Peter Medawar Building for Pathogen Research, University of Oxford, Oxford, UK. [5]Wellcome Centre for Human Genetics, University of Oxford, Oxford, UK. [6]Wellcome Sanger Institute, Hinxton, UK. [7]Human Technopole, Milan, Italy. [8]Big Data Institute, Nuffield Department of Medicine, University of Oxford, Oxford, UK. [9]Department of Biology, University of Oxford, Oxford, UK. [10]The Walter and Eliza Hall Institute of Medical Research, Parkville, VIC, Australia. [11]Department of Medical Biology, University of Melbourne, Melbourne, VIC, Australia. [12]Translational Biomarkers, Infectious Diseases Therapeutic Area, Janssen Pharmaceutica NV, Beerse, Belgium. [13]Department of Pediatrics, Wilhelmina Children's Hospital, University Medical Center Utrecht, Utrecht, Netherlands. [14]ReSViNET Foundation, Zeist, Netherlands. [15]National Heart and Lung Institute, Imperial College London, London, UK. [16]Translational Pediatrics and Infectious Diseases, Pediatrics Department, Hospital Clínico Universitario de Santiago de Compostela, Santiago de Compostela, Spain. [17]Genetics, Vaccines, Infectious Diseases and Pediatrics Research Group (GENVIP), Instituto de Investigación Sanitaria de Santiago, University of Santiago de Compostela, Santiago de Compostela, Spain. [18]Centro de Investigación Biomédica en Red de Enfermedades Respiratorias (CIBERES), Instituto de Salud Carlos III, Madrid, Spain. [19]Centre for Global Health, Usher Institute, Edinburgh Medical School, University of Edinburgh, Edinburgh, UK. [20]MRC/Wits Rural Public Health and Health Transitions Research Unit (Agincourt), School of Public Health, Faculty of Health Sciences, University of the Witwatersrand, Johannesburg, South Africa. [21]Sydney Infectious Diseases Institute, School of Medical Sciences, Faculty of Medicine and Health, University of Sydney, Sydney, Australia. [22]These authors jointly supervised this work: Tanya Golubchik, Andrew J. Pollard. ✉e-mail: gulung.lin.oxford@gmail.com; tanya.golubchik@sydney.edu.au

## RESCEU Consortium

Gu-Lung Lin ®[1,2]✉, Simon B. Drysdale ®[1,2,3], Matthew D. Snape[1,2], Daniel O'Connor ®[1,2], Jeroen Aerssens[12], Louis Bont ®[13,14], Peter J. M. Openshaw ®[15], Federico Martinon-Torres ®[16,17,18], Harish Nair ®[19,20] & Andrew J. Pollard ®[1,2,22]

A full list of members and their affiliations appears in the Supplementary Information.

