## [Peer Review File · Nature Communications]

Targeted metagenomics reveals association between severity and pathogen co-detection in infants with respiratory syncytial virusREVIEWER COMMENTS

Reviewer #1 (Remarks to the Author):

This study used RNA-sequencing to detect co-pathogens in the nasopharynx of infants with RSV bronchiolitis and found that the presence of Haemophilus was associated with disease severity. The strength of the study was the inclusion of participants from multiple institutions during multiple viral seasons. The manuscript could be improved by expanding the discussion about the clinical relevance of the findings, including more information about confounders, and providing data tables summarizing sub analyses. Multiple studies have previously demonstrated an association between Haemophilus and RSV disease severity in infants. The manuscript does not clearly state what this study adds to our knowledge and understanding of the role of co-pathogens in RSV bronchiolitis.

Study Population & Methods

When was the ReSVinet score calculated? Some of the components reflect the entire illness while others are clinical factors that are dynamic and change over time. Is the score reported the mean score for a participant? The peak score? The first score on presentation to care? This affects the interpretation of the overall conclusions of the study as the ReSVinet score was the primary outcome for disease severity.

What proportion of these infants received breastmilk versus formula? As infant feeding practices can impact both the microbiome and the outcomes of respiratory virus infections, this is an important confounder to consider.

Correlates of RSV disease severity

In this section of the results, the authors state in line 177 that infants with viral co-infection were more likely to require intensive care and mechanical ventilation than those without. However, the infants with viral co-detections had a higher proportion of participants with a medical comorbidity. The comorbidities present were not well defined. In the footnote of Figure 1 they are listed as "Comorbidities included prematurity with or without

bronchopulmonary dysplasia, ventricular septal defect, and other congenital abnormalities.” While the presence or absence of a comorbidity was included as a covariate in the analysis, different comorbidities will have different degrees of risk associated with RSV severity. Since the proportion of the study population with a comorbidity is low, this conclusion would be strengthened with a subgroup analysis of only healthy participants.

Beginning at line 178, the authors state that each individual virus was analyzed separately. There is not a table to correspond to these analyses and it should be provided in the supplement for the readers.

Line 182 states there is “some evidence” that infants co-infected with HHV-6 were more likely to require intensive care but the analysis was not significant after adjusting for multiple comparisons. The authors appropriately adjusted to reduce false positive rates and found no significant difference. Thus, there is no evidence to support this statement and it should be removed.

RSV and Co-detected bacterial pathogens

The authors suggest the 27 infants without detection of bacteria was due to poor sample quality. Their explanation is that the RSV reads were lower in this group relative to the rest, suggesting poor quality. These infants were excluded from the analysis of bacterial co-detection but were included in the analysis of viral co-detection. Did any of these samples excluded during this part of the analysis have viral co-detection? If these samples were deemed insufficient, then they should not be included in any of the analysis to remain consistent.

Line 210 discusses a comparison of children with *M. catarrhalis*, *S. pneumoniae*, and *H. influenzae*. This data is not present in any table and should be provided in the supplemental tables.

Positive association between *Haemophilus* presence and RSV severity

The paragraph beginning at line 220 discusses the association of clinical outcomes with the presence of Haemophilus. The authors acknowledge the effect size is small. What is the clinical relevance of these findings? These are not discussed in the discussion section. Further, in supplemental table 10, the average ReSVinet scores between the two groups are 8.5 and 6.8. Those both fall within the moderate distress range (Justicia-Grande AJ, Martinon-Torres F. The ReSVinet Score for Bronchiolitis: A Scale for All Seasons. Am J Perinatol 2019;36(S 02):S48-S53. DOI: 10.1055/s-0039-1691800). Is an absolute change in ReSVinet score of 1.8-2.5 clinically meaningful?

In the birth cohort, did any of the infants have RSV negative samples from prior to their first RSV infection? If so, was did Haemophilus colonization precede RSV infection and was it still associated with disease severity in this subset? This would be similarly interesting for Moraxella.

Negative association between Moraxella presence and RSV severity

In the paragraph beginning with line 248, the authors infer that administration of antibiotics to a participant implies a secondary bacterial infection. This is problematic for several reasons. First, what is the nature of the presumed secondary bacterial infection? Many infants with RSV receive antibiotics simply because of perceived severity of illness. Was there a documented bacterial infection? If so, this information would be important to include as things like acute otitis media, urinary tract infection, superimposed bacterial pneumonia, or bacteremia would alter a participant's respiratory status differently and may or may not be related to the presence of Haemophilus or other bacteria in the nasopharynx. Further, how many doses of antibiotics did these participants receive? Was it a single dose or a full course? Details on the antibiotic choice, duration and clinical indication should be expanded upon or this paragraph should be removed.

Discussion

In line 279 the authors comment on co-detection of viruses. How do the findings of this manuscript compare to what is known in the literature? The discussion comparing the

findings of this study to the literature should be expanded.

The word “as” can be deleted in line 304

Line 310 – There are multiple studies cited (refs 34, 35, 36, 38, 39) as well as others that were not cited (Chu CY, Qiu X, McCall MN, et al. Airway Gene Expression Correlates of Respiratory Syncytial Virus Disease Severity and Microbiome Composition in Infants. *J Infect Dis* 2021;223(9):1639-1649. DOI: 10.1093/infdis/jiaa576) which have the same finding as this study: that the presence of Haemophilus is associated with more severe disease in infants with RSV bronchiolitis. The authors do not highlight what is different, new, or novel about the findings in this study compared to those done in the past. What does this study add to our knowledge and understanding of co-detection of Haemophilus or other pathogens during RSV bronchiolitis?

In line 316 the authors cite a study that contradicted their finding of no association between streptococcus abundance and need for hospitalization. Why might there be a difference in findings between these two studies? This should be discussed.

Line 317. As above, information about the diagnosis of superimposed bacterial infections and duration/type of antibiotics should be expanded upon or removed as there is insufficient information to discern clinical concern for superimposed bacterial infections that would alter clinical course.

Reviewer #2 (Remarks to the Author):

I co-reviewed this manuscript with one of the reviewers who provided the listed reports. This is part of the Nature Communications initiative to facilitate training in peer review and to provide appropriate recognition for Early Career Researchers who co-review manuscripts

Reviewer #3 (Remarks to the Author):

Lin et al present their work on targeted metagenomics data from RSV-positive pediatric

patients. Their main focus is to report co-infections with viruses and three bacteria and examine associations with severity. Lin et al have used a respiratory panel to generate the metagenomics data, and this is the first time RSV patients' samples have been examined this way, as in most studies in the literature, co-infections are investigated via PCR. Overall, the sample size of patients is good, the analysis is sound and the results are in line with current knowledge. The main result is the generation of the sequencing data and this is really an association-only study and the associations discussed have already been discussed elsewhere, albeit in smaller cohorts. It would be worthwhile to include other data if available, from

- 1) other non-RSV respiratory paediatric patients as controls/different category so that associations with severity and interaction between co-infection pathogens can be assessed in a more informed manner.
- 2) Characterisation of the microbial community for RSV-patients versus controls.

I have some specific criticisms, see below.

1) A main limitation of the study IMO is the arbitrary nature of deciding which virus is present. In supplementary Table 12, the criteria for considering presence of absence are different for every single virus, without an explanation. How were these thresholds decided, on what data/basis? I would understand if DNA and RNA viruses had different thresholds, but even CMV and HHV6 differ between them - and similarly other RNA viruses.

2) In Supplementary Table 8, it is stated: "These genera were not included in the Castanet enrichment panel, so their read numbers were expected to be lower as these were detected by unenriched metagenomics only."

There was no mention of unenriched metagenomics in any part of the report? Were the results consistent between targeted & non targeted metagenomics? Were there any viruses not covered in the probeset uncovered by non-targeted CMg? IMO either a lot more information should be included on the comparison/findings between the two metagenomics approaches, or the bacteria found by untargeted metagenomics should not be reported in this paper, given also there is no discussion about these findings.

3) As far as I am aware this is the first report using targeted metagenomics to characterise RSV-positive samples and on cohorts of this size. The established literature is on smaller cohorts and they examine positivity via virus PCR testing usually, with only few papers on untargeted metagenomics.

There are numerous papers discussing association with severity (or lack thereof) that are not discussed and should be, as this is not the first ever paper addressing this question.

Some examples of overlooked papers

<https://erj.ersjournals.com/content/42/2/461>

<https://www.ncbi.nlm.nih.gov/pmc/articles/PMC7295447/#R7>

<https://pubmed.ncbi.nlm.nih.gov/20700753/>

<https://onlinelibrary.wiley.com/doi/full/10.1002/jmv.28753>

4) The data collected and the analyses reported seem valid as described, however in some instances further details are required. For example, how were the metagenomics data analyses for species identification? Was a taxonomic classification method used or did the authors choose the X pathogens they were interested in? Were there any viral/bacterial reads in the negative controls and how were the negative controls used? Presence/absence or ratio of reads/RPM above a threshold? More details needed.

5) The actual R code for the statistical analyses (regressions etc) is not provided, would be good to provide for reproducibility.

6) Sometimes the p-values reported are borderline significant and it would be good to acknowledge this explicitly in these cases.

7) It is stated that 7 infants from the RSV-positive cases were excluded, can the authors explain why the viral load was undetected while being PCR positive? Is it relevant to limits of detection of the targeted metagenomics or possibly quality of clinical samples?

8) For the 9 participants that had both RSV types, was this a true mixed infection, contamination or analytical artefact?

9) Figure 3: Why are the results from the longitudinal birth cohort study and the infant cross-sectional study are shown separately for Haemophilus species but not the other 2 bacterial species?

Reviewer #4 (Remarks to the Author):

This study is a large scale investigation of the potential impact of co-occurring microorganisms on the clinical severity of RSV infections in infants. Using targeted metagenomics, phylogenetic, and statistical analysis the researchers were able to determine that there were 3 bacterial genera with a significant impact, negative and positive correlations on RSV severity.

My impression of this work is positive and I recommend publication with minor revisions. I have addressed these specific suggestions below:

Minor revisions

- The introductory sentences of paragraph 2 in the introduction are confusing: the treatment prevents severe disease such as RSV caused pneumonias. "Palivizumab, a monoclonal antibody for RSV prevention" sounds as if this is a vaccination instead of prophylaxis.

- 'Co-occurrence' would be a better term when the presence of a virus is accounted for but replication (determined by RT-qPCR, etc) and active infection was not determined

- would like to see the background info on ResVinet score in the introduction especially as

the topic sentence of an intro paragraph regards supportive care, the score is introduced previously without the details that appear later in the results section, would like to see addressed for clarity and narrative cohesion

o a very brief description of how the scoring is assigned along with the total points possible would be useful in help understand the point increase results for haemophilus & RSV, would suggest moving from methods to introduction

o

- “ This was because infants in the infant cross-sectional study were recruited on presentation with an RSV infection, while those in the longitudinal birth cohort study were previously healthy and prospectively followed up to detect RSV positivity and thus more likely to include mild cases. The association with Haemophilus was robust to the difference in study population between the cohorts, supporting the separate effect of Haemophilus to known effects of age.” Move explanation to discussion

- Would like to see the antibiotic use impact hypothesis addressed earlier on in paper, it’s a common factor to look at when investigating co-occurring bacteria

- While a deeper look at the impact of applied antibiotics on the different genera is out of the scope of this paper, a mention of this in the discussion would be useful when considering the difference between a potentially protective bacteria and a co-occurring harmful one (especially as one is more susceptible to 1st and 2nd gen cephalosporins and one is not)

- “Co-detected viruses could represent either active replication of an additional virus (either newly acquired or reactivated) or leftover nucleic acid from a previous infection; the additional virus may be either as a non-pathogenic ‘bystander’ or a true disease-causing agent.” – suggest rewriting for clarity

- “Although not definite, unique read count as a surrogate for viral load may provide a clue to differentiate between these possibilities. Co-detected viruses with low viral load may come from prolonged shedding from a previous infection, whereas those with high viral load are more likely to cause symptoms. In this study, we focused on viruses with evidence

of substantial viral load based on read numbers, as being consistent with active replication and therefore possible co-infection.” Would like to see citations for this paragraph.

Overall, this is a well-structured paper. A few minor edits will improve readability. The discussion lacks potential explanation for the phylogenetic analysis and subsequent result in any detail. The supplemental phylogenetic analysis indicates a large analysis was performed. The bioinformatic methods section was well written and easy to follow, which will be appreciated by anyone replicating or applying it.

RESPONSE TO REVIEWER COMMENTS

Reviewer #1 (Remarks to the Author):

This study used RNA-sequencing to detect co-pathogens in the nasopharynx of infants with RSV bronchiolitis and found that the presence of *Haemophilus* was associated with disease severity. The strength of the study was the inclusion of participants from multiple institutions during multiple viral seasons. The manuscript could be improved by expanding the discussion about the clinical relevance of the findings, including more information about confounders, and providing data tables summarizing sub analyses. Multiple studies have previously demonstrated an association between *Haemophilus* and RSV disease severity in infants. The manuscript does not clearly state what this study adds to our knowledge and understanding of the role of co-pathogens in RSV bronchiolitis.

We express our gratitude to the reviewer for the review of our manuscript and the constructive suggestions to improve our work. Please see below our point-by-point responses in blue. All line numbers included in the responses refer to the manuscript with tracked changes.

Study Population & Methods

When was the ReSVinet score calculated? Some of the components reflect the entire illness while others are clinical factors that are dynamic and change over time. Is the score reported the mean score for a participant? The peak score? The first score on presentation to care? This affects the interpretation of the overall conclusions of the study as the ReSVinet score was the primary outcome for disease severity.

The ReSVinet score was calculated upon presentation for care during the acute RSV infection. This means that the ReSVinet score was determined at the time when a participant in the longitudinal birth cohort study was visited by the study staff and tested positive for RSV, or when a participant in the infant cross-sectional study was enrolled. This information has been added to the manuscript (Lines 516–518).

We acknowledge that there is a limitation associated with this score since it changes over time. Therefore, we also assessed the associations with individual clinical variables (e.g., requirement for hospitalisation, intensive care, etc.) to account for this limitation.

What proportion of these infants received breastmilk versus formula? As infant feeding practices can impact both the microbiome and the outcomes of respiratory virus infections, this is an important confounder to consider.

Thank you for pointing this out. We agree that breastfeeding status can affect both the microbiome and the outcomes of respiratory tract infection. We collected this information in the longitudinal birth cohort study when the participants were followed up at the age of 1 year; however, not all participants

had this information available. Breastfeeding status was not collected in the infant cross-sectional study.

Overall, 12% (51/433) of the infants had information available about their breastfeeding history. Among them, 84% (43/51) had been breastfed during the first year of life. Taken into account the duration of breastfeeding, 53% (27/51) had been breastfed within the 4 weeks prior to the RSV infection, either exclusively (N= 21) or in combination with formula milk (N = 6).

Demographic, virological, and clinical features were comparable between infants who were breastfed within the 4 weeks leading up to the RSV infection and those who were not (Supplementary Table 13). At least one bacterial genus was found in 93% (25/27) of the breastfed infants (either exclusively or in combination of formula milk) and 100% of the exclusively formula-fed infants within the 4 weeks prior to the RSV infection. A mean of 2.1 and 2.5 bacterial genera were identified in each breastfed and formula-fed infants, respectively, and this difference was not significant (two-tailed Mann–Whitney U test, $P = 0.115$). These genera shared similar frequencies in the two groups, with *Moraxella*, *Streptococcus*, and *Haemophilus* being the most frequently recovered genera (Supplementary Fig. 10). *Granulicatella*, *Lactococcus*, and *Gemella* spp. were only found in breast-fed infants (however, they were not the targeted species in the *Castanet* enrichment panel, and the broader infant microbiome is outside the scope of the present study).

We have included these results in main text (Lines 265–268), Supplementary Table 13, and Supplementary Fig. 10.

Correlates of RSV Disease Severity

In this section of the results, the authors state in line 177 that infants with viral co-infection were more likely to require intensive care and mechanical ventilation than those without. However, the infants with viral co-detections had a higher proportion of participants with a medical comorbidity. The comorbidities present were not well defined. In the footnote of Figure 1 (*this should be Table 1*) they are listed as “Comorbidities included prematurity with or without bronchopulmonary dysplasia, ventricular septal defect, and other congenital abnormalities.” While the presence or absence of a comorbidity was included as a covariate in the analysis, different comorbidities will have different degrees of risk associated with RSV severity. Since the proportion of the study population with a comorbidity is low, this conclusion would be strengthened with a subgroup analysis of only healthy participants.

Thank you for the suggestion. The covariates with the lowest Akaike information criterion (AIC) included in the general linear models for PICU admission and mechanical ventilation were age, gestational age, RSV subgroup, sampling country, study cohort, peak RSV read count, and duration between symptom onset and sampling. Comorbidity was not selected based on the AIC, suggesting it was not independently predictive of either outcome. We note that prematurity accounted for the majority of comorbidities in our study population (24/35, 69%), and gestational age was a significant covariate in both models ($P = 0.002$ and 7.7×10^{-5} , respectively). Therefore, the effects of different comorbidities were largely accounted for by gestational age; the lower the gestational age, the higher

the rates of PICU admission and mechanical ventilation. Since the models already included gestational age as a covariate and comorbidity was not independently predictive of the outcomes, we would consider that these models have sufficiently explained the effects of all potential confounders and a subgroup analysis would not be necessary.

The information about both confounders (i.e., gestational age and comorbidity) has been included in the manuscript (Lines 226–229)

Beginning at line 178, the authors state that each individual virus was analyzed separately. There is not a table to correspond to these analyses and it should be provided in the supplement for the readers.

Thank you for the suggestion. We have included all the analyses performed on each additional virus family in Supplementary Tables 7–10.

Line 182 states there is “some evidence” that infants co-infected with HHV-6 were more likely to require intensive care but the analysis was not significant after adjusting for multiple comparisons. The authors appropriately adjusted to reduce false positive rates and found no significant difference. Thus, there is no evidence to support this statement and it should be removed.

Thank you. We have removed ‘some evidence’ and amended this sentence accordingly (Lines 234–239).

RSV and Co-Detected Bacterial Pathogens

The authors suggest the 27 infants without detection of bacteria was due to poor sample quality. Their explanation is that the RSV reads were lower in this group relative to the rest, suggesting poor quality. These infants were excluded from the analysis of bacterial co-detection but were included in the analysis of viral co-detection. Did any of these samples excluded during this part of the analysis have viral co-detection? If these samples were deemed insufficient, then they should not be included in any of the analysis to remain consistent.

Apologies for the confusion, but these 27 infants were not excluded from the analyses of associations between bacterial co-detection and clinical outcomes. These 27 infants were only excluded in Line 257 where we presented the median number of targeted bacterial species found in the remaining infants. We have amended this sentence and reported the descriptive statistics in all 433 infants to avoid any confusion (Lines 257–258).

Regarding the status of viral co-detection in these 27 infants, 18 had no viral co-detection, eight had viral co-detection, and one had equivocal viral co-detection.

Line 210 (*Line 264 in the revised manuscript with tracked changes*) discusses a comparison of children with *M. catarrhalis*, *S. pneumoniae*, and *H. influenzae*. This data is not present in any table and should be provided in the supplemental tables.

Thank you for the suggestion. To clarify, Lines 262–264 discuss a comparison between infants with any of these three bacterial **genera** (i.e., *Moraxella*, *Streptococcus*, and *Haemophilus*) and those without. It is not just the three bacterial **species** (i.e., *M. catarrhalis*, *S. pneumoniae*, and *H. influenzae*). In addition, we had provided descriptive statistics (median and interquartile range) and the statistical test (Mann–Whitney U test) in the text.

To address your comment, we have included a box and whisker plot with overlaid data points for visualisation of the data (Supplementary Fig. 9).

Positive Association Between *Haemophilus* Presence and RSV Severity

The paragraph beginning at line 220 discusses the association of clinical outcomes with the presence of *Haemophilus*. The authors acknowledge the effect size is small. What is the clinical relevance of these findings? These are not discussed in the discussion section. Further, in supplemental table 10, the average ReSVinet scores between the two groups are 8.5 and 6.8. Those both fall within the moderate distress range (Justicia-Grande AJ, Martinon-Torres F. The ReSVinet Score for Bronchiolitis: A Scale for All Seasons. Am J Perinatol 2019;36(S 02):S48-S53. DOI: 10.1055/s-0039-1691800). Is an absolute change in ReSVinet score of 1.8-2.5 clinically meaningful?

We agree that interpreting the clinical significance of the difference in ReSVinet scores between 8.5 ± 4.5 and 6.8 ± 4.6 may be challenging. However, when looking at the severity group based on the ReSVinet score (0–7 vs. 8–13 vs. 14–20) and compared with infants without *Haemophilus*, a higher proportion of infants with *Haemophilus* had moderate RSV disease (ReSVinet scores 8–13; 36% vs. 23%), while a lower proportion of infants with *Haemophilus* had mild RSV disease (ReSVinet scores 0–7; 48% vs. 65%). The proportions of infants with severe RSV disease were similar in both groups (ReSVinet scores 14–20; 16% vs. 12%). In addition, we showed that infants with *Haemophilus* were significantly more likely to have a fever and require hospitalisation and respiratory support, including mechanical ventilation, than those without. All these outcome variables are clinically relevant and meaningful, associated with disease severity and patient care. We have included a discussion regarding the clinical relevance of these findings in the Discussion section (Lines 398–401).

In the birth cohort, did any of the infants have RSV negative samples from prior to their first RSV infection? If so, was did *Haemophilus* colonization precede RSV infection and was it still associated with disease severity in this subset? This would be similarly interesting for *Moraxella*.

Unfortunately, we do not have data to address this question. In our study, we only sequenced samples testing positive for RSV by either point-of-care testing or the hospital’s antigen and PCR assays, along with very few RSV-negative samples collected from infants with respiratory symptoms as negative controls.

Negative Association Between *Moraxella* Presence and RSV Severity

In the paragraph beginning with line 248, the authors infer that administration of antibiotics to a participant implies a secondary bacterial infection. This is problematic for several reasons. First, what

is the nature of the presumed secondary bacterial infection? Many infants with RSV receive antibiotics simply because of perceived severity of illness. Was there a documented bacterial infection? If so, this information would be important to include as things like acute otitis media, urinary tract infection, superimposed bacterial pneumonia, or bacteremia would alter a participant’s respiratory status differently and may or may not be related to the presence of *Haemophilus* or other bacteria in the nasopharynx. Further, how many doses of antibiotics did these participants receive? Was it a single dose or a full course? Details on the antibiotic choice, duration and clinical indication should be expanded upon or this paragraph should be removed.

Among the 433 infants, 361 had available information on antibiotic usage (91 received antibiotics and 270 did not). Among the 91 infants who received antibiotics, bacterial infection was suspected in 88 infants (this information was not available in two infants). Among the 88 infants who received antibiotics and were suspected to have a bacterial infection, 30 had documented bacterial isolates clinically, including four from blood samples, 27 from bronchial wash samples, one from a nasopharyngeal swab, and one without available information about the sample type (two infants had bacteria isolated from both blood and bronchial wash, and one infant had bacteria isolated from both bronchial wash and a nasopharyngeal swab). Information on the timing of sampling of these clinical samples was not available. Note, all infants with documented bacterial isolates received antibiotics.

The bacterial isolates from either bronchial wash or a nasopharyngeal swab, along with the percentage of those recovered by our targeted metagenomic sequencing, are shown in the table below.

Clinical bacterial isolates	Frequency	Recovery rate of targeted metagenomics
Staphylococcus aureus	17	65%
Haemophilus influenzae	9	100%
Moraxella or M. catarrhalis	9	67%
Streptococcus pneumoniae	6	83%
Klebsiella pneumoniae	2	0%
Escherichia coli	1	0%
Streptococcus pyogenes	1	0%
Streptococcus agalactiae	1	100%
Enterobacter cloacae complex	1	100%

This table has been added to the Supplementary Information (Supplementary Table 16).

Regarding the information about antibiotic usage, only the start date, antibiotic name, and route of administration of the antibiotic were available, so we did not know how many doses were given and whether a full course of antibiotics was completed. Since the information was missing and given the complexity the reviewer pointed out, we have revised this section (Lines 301–306), removed the implication of a secondary bacterial infection in the Discussion (Lines 424–425), and added this limitation to the Discussion (Lines 463–467).

Discussion

In line 279 the authors comment on co-detection of viruses. How do the findings of this manuscript compare to what is known in the literature? The discussion comparing the findings of this study to the literature should be expanded.

Thank you for the comments. We have broadened the discussion and placed our findings in the context of existing literature (Lines 381–394). We also discussed the clinical significance and added value of our results (Lines 444–452).

The word “as” can be deleted in line 304.

Thank you, the word ‘as’ has been removed (Line 374).

Line 310 – There are multiple studies cited (refs 34, 35, 36, 38, 39) as well as others that were not cited (Chu CY, Qiu X, McCall MN, et al. Airway Gene Expression Correlates of Respiratory Syncytial Virus Disease Severity and Microbiome Composition in Infants. *J Infect Dis* 2021;223(9):1639-1649. DOI: 10.1093/infdis/jiaa576) which have the same finding as this study: that the presence of *Haemophilus* is associated with more severe disease in infants with RSV bronchiolitis. The authors do not highlight what is different, new, or novel about the findings in this study compared to those done in the past. What does this study add to our knowledge and understanding of co-detection of *Haemophilus* or other pathogens during RSV bronchiolitis?

Thank you for the comments. We have included the citation you mentioned along with other studies (Lines 401–406) and expanded the discussion about the unique aspects and added value of this study (Lines 445–452).

In line 316 the authors cite a study that contradicted their finding of no association between streptococcus abundance and need for hospitalization. Why might there be a difference in findings between these two studies? This should be discussed.

Thank you. We have discussed possible explanations for the differences in findings between our study and others (Lines 444–445).

Line 317. As above, information about the diagnosis of superimposed bacterial infections and duration/type of antibiotics should be expanded upon or removed as there is insufficient information to discern clinical concern for superimposed bacterial infections that would alter clinical course.

Thank you for the suggestion. As mentioned in our response to your earlier comment on antibiotics, we have removed this section in Discussion (Lines 424–427).

Reviewer #2 (Remarks to the Author):

We are grateful to the reviewer for their support in the peer review of our work.

Reviewer #3 (Remarks to the Author):

Lin et al present their work on targeted metagenomics data from RSV-positive pediatric patients. Their main focus is to report co-infections with viruses and three bacteria and examine associations with severity. Lin et al have used a respiratory panel to generate the metagenomics data, and this is the first time RSV patients' samples have been examined this way, as in most studies in the literature, co-infections are investigated via PCR. Overall, the sample size of patients is good, the analysis is sound and the results are in line with current knowledge. The main result is the generation of the sequencing data and this is really an association-only study and the associations discussed have already been discussed elsewhere, albeit in smaller cohorts. It would be worthwhile to include other data if available, from

- 1) Other non-RSV respiratory paediatric patients as controls/different category so that associations with severity and interaction between co-infection pathogens can be assessed in a more informed manner.
- 2) Characterisation of the microbial community for RSV-patients versus controls.

We express our gratitude to the reviewer for the review of our manuscript and the constructive suggestions to improve our work. Please see below our point-by-point responses in blue. All line numbers included in the responses refer to the manuscript with tracked changes.

We acknowledge that comparisons of the microbial community and its association with disease severity between RSV-positive and RSV-negative patients offer additional insight into the characteristics of patients with respiratory infection. However, in this study, our aim was to characterise the microbial associations with severity specifically in RSV-positive infants, using an extremely broad and specific method that is able to resolve genetic sequences of over 100 targeted pathogens (distinct from detection-only methods such as PCR), at a scale that has not been reported before. Three bacterial genera were most abundant in this cohort (*Streptococcus*, *Moraxella*, *Haemophilus*), but all samples were screened for the full complement of targeted organisms, including 26 bacterial genera. In line with our aims, we did not calibrate disease severity against the background population risk or RSV-negative infants, as we sequenced only RSV-positive samples (with very few RSV-negative samples as negative controls). Therefore, we did not and could not perform analysis in RSV-negative infants using the current dataset.

I have some specific criticisms, see below.

1) A main limitation of the study IMO is the arbitrary nature of deciding which virus is present. In supplementary Table 12, the criteria for considering presence of absence are different for every single virus, without an explanation. How were these thresholds decided, on what data/basis? I would understand if DNA and RNA viruses had different thresholds, but even CMV and HHV6 differ between them - and similarly other RNA viruses.

Thank you for your comment and question. We established the thresholds for determining the presence and absence of each virus on the basis of the number of unique (deduplicated) reads mapped to the reference genomes and the proportion of the genome recovered. However, we agree with the reviewer that a single standardised threshold is preferable.

Accordingly, we have updated and standardised the thresholds for all viruses based solely on the proportion of the genome recovered. We dropped the use of read count to determine the presence status because this metric is more sensitive to batch effects in capture efficiency, and is also affected by the length of the genome. The updated thresholds are uniformly >35% as present, 15–35% as equivocal, and <15% as absent. We thank the reviewer for the suggestion, which we believe has substantially improved the manuscript.

As an additional sensitivity check, we cross-checked detections using taxonomic classification data generated from Kraken 2, a metagenomic sequence classification tool, using a library containing all viral and bacterial RefSeq genomes, along with the human genome. All viruses that were identified as present based on the threshold criteria above had corresponding reads classified as the same virus by Kraken 2, with the exception of enteroviruses. This is because Kraken 2 has a very limited enterovirus library and thus is not sensitive to this genetically diverse virus. The *Castanet* pipeline was significantly better at identifying reads classified as enterovirus relative to Kraken 2 (mean read count: 25,872 vs. 9,603).

We have updated the Methods section to reflect the updated standardised thresholds (Lines 600–608) and removed original Supplementary Table 12. We are pleased to report that updating these thresholds had a minimal effect on the presence/absence status of co-detected viruses, with changes in 77 out of 2,056 hits (3.7%) across all viral organisms. All the association analyses and results have been updated accordingly. No major changes to the results were noted.

2) In Supplementary Table 8 (*Supplementary Table 12 in the revised Supplementary Information*), it is stated: “These genera were not included in the *Castanet* enrichment panel, so their read numbers were expected to be lower as these were detected by unenriched metagenomics only.”

There was no mention of unenriched metagenomics in any part of the report? Were the results consistent between targeted & non targeted metagenomics? Were there any viruses not covered in the probeset uncovered by non-targeted CMg? IMO either a lot more information should be included on the comparison/findings between the two metagenomics approaches, or the bacteria found by untargeted metagenomics should not be reported in this paper, given also there is no discussion about these findings.

We apologise for the confusion. We did not conduct separate unenriched metagenomics. Instead, with *Castanet*, as with all capture-based protocols, targeted sequences are selectively amplified by 100–1000x during library preparation, but untargeted sequences remain at a lower level (commonly referred to as “metagenomic background” in this context) and are subsequently sequenced alongside the targeted ones. We used the PubMLST multi-species isolate database to identify matching bacteria, thus allowing us to classify contig sequences (assembled from the sequencing reads) to a certain taxonomic level, even though they were not part of the *Castanet* panel.

We added this explanation of metagenomic background to the footnote of Supplementary Table 12.

3) As far as I am aware this is the first report using targeted metagenomics to characterise RSV-positive samples and on cohorts of this size. The established literature is on smaller cohorts and they examine positivity via virus PCR testing usually, with only few papers on untargeted metagenomics.

There are numerous papers discussing association with severity (or lack thereof) that are not discussed and should be, as this is not the first ever paper addressing this question. Some examples of overlooked papers

<https://erj.ersjournals.com/content/42/2/461>

<https://www.ncbi.nlm.nih.gov/pmc/articles/PMC7295447/#R7>

<https://pubmed.ncbi.nlm.nih.gov/20700753/>

<https://onlinelibrary.wiley.com/doi/full/10.1002/jmv.28753>

Thank you for the comment and acknowledgment of the novelty of this study. We have expanded the discussion and included the literature you mentioned along with other studies (Lines 346–347, 385–393).

4) The data collected and the analyses reported seem valid as described, however in some instances further details are required. For example, how were the metagenomics data analyses for species identification? Was a taxonomic classification method used or did the authors choose the X pathogens they were interested in? Were there any viral/bacterial reads in the negative controls and how were the negative controls used? Presence/absence or ratio of reads/RPM above a threshold? More details needed.

Thank you for the comments and questions. We described the methods of species identification in the Methods section, specifically in “Nucleic acid isolation and targeted metagenomic sequencing,” “Respiratory viral genome reconstruction,” and “Respiratory bacterial reconstruction.” The species included in the probe set were chosen due to their potential for pathogenicity and covered over 100 common human bacterial and viral pathogens (Supplementary Table 17).

For virus identification from the sequencing reads, we used the *Castanet* pipeline, which mapped the sequencing reads to all the reference genomes that were used to design the probe set. The coverage of all reconstructed genomes, either complete or partial, were subsequently used to determine the presence status of each targeted virus. Kraken 2 was also used to cross check the detections. Therefore,

we were able to detect and identify all the common viral pathogens included in the probe set; no pathogen selection was done in this step.

Regarding bacterial identification, we used the “Species ID” functionality in the PubMLST multi-species isolate database to search for matching bacteria among contig sequences (assembled by metaSPAdes from the sequencing reads). The PubMLST database integrates genome information of over 130 different microbial species and genera with extensive genome taxonomic coverage for bacterial identification and ribosomal multilocus sequence typing (rMLST). Similarly, no additional pathogen selection was done in this step.

In terms of negative controls, we had two types of negative controls included in the sequencing assays: (1) no template controls which were pure M4RT[®] transport medium, and (2) RSV-negative controls collected from participants who had respiratory symptoms but tested negative for RSV by either point-of-care testing or the hospital’s antigen or PCR assays. These negative controls were processed alongside RSV-positive samples from nucleic acid extraction to library preparation and sequencing; therefore, the no template controls were true extraction controls. We did not sequence samples collected from healthy infants. This information has been included in the Methods (Lines 549–555).

Among the nine extraction controls (buffer only, taken through extraction alongside real samples), none had reads mapped to any of the reference genomes of viruses and bacteria of interest. The only exception was that *Burkholderia multivorans* was present in three of the extraction controls based on the criteria we implemented. This is presumably a result of kit contaminants (i.e., kitome), commonly observed in large-scale sequencing studies. This organism was not targeted by *Castanet* as it is not considered a species of interest, and was not included in any of the association analyses. We have added the information about extraction controls to the Results (Lines 148–152).

Among the 11 RSV-negative respiratory swab controls, all had at least one viral or bacterial species identified. The most common viruses were rhinovirus (N = 6) and human herpesvirus 6 (N = 2), followed by human adenovirus, human coronavirus HKU1, influenza A virus, and human parainfluenza virus (one each). The most common bacteria were *Moraxella* spp. (N = 13, including *M. catarrhalis*, *M. nonliquefaciens*, and *M. lincolnii*) and *Haemophilus influenzae* (N = 3), followed by *Dolosigranulum pigrum*, *Streptococcus mitis*, and *Streptococcus pneumoniae* (one each). These detections of respiratory pathogens are unsurprising, as the RSV-negative swab controls were derived from patients who had respiratory symptoms that were clinically indistinguishable from RSV. This information has been added in Lines 152–159.

5) The actual R code for the statistical analyses (regressions etc) is not provided, would be good to provide for reproducibility.

The R code used for the descriptive statistics and statistical analyses in each table is available in Supplementary Data 2.

6) Sometimes the p-values reported are borderline significant and it would be good to acknowledge this explicitly in these cases.

The exact P values have been provided in the text, tables, and figures to help the readers determine the significance. Except for the difference in age between infants with co-detection of human coronavirus and RSV and those with RSV alone ($P = 0.046$; Supplementary Fig. 4), all other reported P values and Q values were ≤ 0.04 and ≤ 0.03 , respectively. Therefore, we consider all these to be significant, instead of borderline significant.

7) It is stated that 7 infants from the RSV-positive cases were excluded, can the authors explain why the viral load was undetected while being PCR positive? Is it relevant to limits of detection of the targeted metagenomics or possibly quality of clinical samples?

We would like to clarify the distinction between initial diagnostic tests (point-of-care testing and the hospital's antigen or PCR assays), used to identify RSV positives, and RT-qPCR, which we used to quantify RSV viral load in a subset of samples. There were seven infants whose samples were positive on initial diagnostic tests but yielded zero RSV sequencing reads. Among the seven infants, five were also negative on re-testing by RT-qPCR. Sequencing and RT-qPCR were done on the same sample, but these samples were distinct from the initial diagnostic swab. Therefore, the second swab may have been of poorer quality, or the first was a false positive; in either scenario, our sequencing and RT-qPCR were in agreement. Of remaining 2/7 samples, one had no RT-qPCR data and we therefore cannot verify its PCR positivity, while the final sample had a substantial viral load (18,463 copies/mL), which falls well within our sequenceable range (we generated RSV reads from 64/93 [69%] of samples with a viral load below 18,463 copies/mL). This suggests a sporadic failure, likely at the library preparation stage.

This information was added to the main text (Lines 137–142).

8) For the 9 participants that had both RSV types, was this a true mixed infection, contamination or analytical artefact?

Thank you for the good question. Prompted by this question, we explored the potential RSV co-infections further. Firstly, we identified the presence of both RSV subgroups in samples by inspecting the contig sequences aligned with the RSV reference genomes. These contig sequences were assembled from the sequencing reads using metaSPAdes or IVA and then aligned with the reference genomes using shiver. In these nine samples, two distinct sequence patterns were observed among the aligned contig sequences, representing two subgroups. Hence, it is unlikely that this phenomenon was attributable to an analytical artefact. However, definitively categorising these occurrences as a true mixed infection or contamination remains challenging. Among the 3/9 samples that displayed good genome coverage ($>50\%$) for both subgroups, none of the genome sequences were identical or nearly identical to those recovered from other infants within the same sequencing batch. In addition, one of the three samples tested positive for both subgroups by RT-qPCR with a high viral load, 2.1×10^7 copies/mL for RSV-A and 2.2×10^7 copies/mL for RSV-B. These three samples were likely to represent genuine mixed infections. In contrast, the remaining 6/9 samples exhibited poor genome coverage ($<15\%$) for one of the subgroups, making it difficult to differentiate between a true mixed infection and contamination on the basis of the sequence similarity of short genome segments.

We did find one case of plausible contamination: a nearly complete RSV-A genome and ~9% of an RSV-B genome were recovered from one of the nine samples. This partial RSV-B genome segment was nearly identical to part of the RSV-B genome recovered from another sample collected in the same season but a different country with a higher viral load (3.1×10^4 vs. 1.6×10^8 copies/mL). These two samples were located adjacent on the sequencing plate. Therefore, it is likely that the identification of RSV-B in this particular sample was due to contamination. We have added this information to the footnote of Supplementary Table 3.

Regarding the other samples that only had short genome segments recovered for one of the subgroups, they either showed dissimilarity to those obtained from other infants within the same sequencing batch or were nearly identical to those recovered from infants within close temporal and geographical proximity (e.g., collected from the same site and within the same week), which would be consistent with epidemiological linkage and cannot be distinguished from contamination.

9) Figure 3: Why are the results from the longitudinal birth cohort study and the infant cross-sectional study are shown separately for *Haemophilus* species but not the other 2 bacterial species?

We appreciate your question. The correlation between ReSVinet score and age is presented for each individual study for *Haemophilus* spp. because the differences between the regression lines in both studies were statistically significant. However, the differences between the regression lines in the individual studies for either *Moraxella* or *Streptococcus* spp. were not significant, except for *Moraxella* spp. in the infant cross-sectional study.

We have added figures for the individual studies for *Moraxella* and *Streptococcus* spp. to Fig. 4 to illustrate the results and to address your question.

Reviewer #4 (Remarks to the Author):

This study is a large scale investigation of the potential impact of co-occurring microorganisms on the clinical severity of RSV infections in infants. Using targeted metagenomics, phylogenetic, and statistical analysis the researchers were able to determine that there were 3 bacterial genera with a significant impact, negative and positive correlations on RSV severity.

My impression of this work is positive and I recommend publication with minor revisions. I have addressed these specific suggestions below:

We appreciate the positive feedback from the reviewer and the comments that help improve our manuscript. Please see below our point-by-point responses in blue. All line numbers included in the responses refer to the manuscript with tracked changes.

Minor revisions

- The introductory sentences of paragraph 2 in the introduction are confusing: the treatment prevents severe disease such as RSV caused pneumonias. “Palivizumab, a monoclonal antibody for RSV prevention” sounds as if this is a vaccination instead of prophylaxis.

Thank you for the comment. We have revised this sentence to avoid any confusion. The sentence now reads “To date, the standard of care for RSV infection has been supportive management. No safe and effective antivirals are available for RSV treatment. Palivizumab, a monoclonal antibody for short-term RSV prophylaxis, has been used in young children at high risk for severe RSV LRTI...” (Lines 67–69).

- ‘Co-occurrence’ would be a better term when the presence of a virus is accounted for but replication (determined by RT-qPCR, etc) and active infection was not determined.

Thank you for the suggestion. We have changed ‘co-infection’ to ‘co-detection’ throughout the manuscript. However, we hypothesised that these co-detected viruses were possible co-infections in the Discussion since there was evidence of substantial viral load based on read numbers, consistent with active viral replication (Lines 378–380).

- would like to see the background info on ReSVinet score in the introduction especially as the topic sentence of an intro paragraph regards supportive care, the score is introduced previously without the details that appear later in the results section, would like to see addressed for clarity and narrative cohesion.

Thank you for the comment. We have moved the description of the ReSVinet score from Methods and Results to Introduction and added more background information to Introduction to improve clarity and cohesion of the manuscript (Lines 75–81).

- a very brief description of how the scoring is assigned along with the total points possible would be useful in help understand the point increase results for *Haemophilus* & RSV, would suggest moving from methods to introduction

Thank you. We have addressed this comment as above (Lines 75–81) and specified in the Methods that the ReSVinet score was calculated by the study staff when the infants were first seen for the acute RSV infection (Lines 516–518).

- “This was because infants in the infant cross-sectional study were recruited on presentation with an RSV infection, while those in the longitudinal birth cohort study were previously healthy and prospectively followed up to detect RSV positivity and thus more likely to include mild cases. The association with *Haemophilus* was robust to the difference in study population between the cohorts, supporting the separate effect of *Haemophilus* to known effects of age.” Move explanation to discussion

Thank you for the suggestion. We have removed the explanation from the Results (Lines 296–298). Since the difference in disease severity between the two study cohorts has been mentioned in the

beginning of the Results (Lines 112–117, 126–127), we have revised the sentence to “As expected on the basis of the difference in study design, ReSVinet scores negatively correlated with age in the infants enrolled in the infant cross-sectional study... but not in those in the longitudinal birth cohort study...” (Lines 292–295).

- Would like to see the antibiotic use impact hypothesis addressed earlier on in paper, it’s a common factor to look at when investigating co-occurring bacteria

Thank you for your comment. We have moved the section about antibiotic use to an earlier section (from Lines 316–323 to Lines 301–306). To address the feedback from Reviewer #1, we have also revised this section since we do not have complete information on antibiotic use.

- While a deeper look at the impact of applied antibiotics on the different genera is out of the scope of this paper, a mention of this in the discussion would be useful when considering the difference between a potentially protective bacteria and a co-occurring harmful one (especially as one is more susceptible to 1st and 2nd gen cephalosporins and one is not)

Thank you. As mentioned in the response above, we have incorporated discussions on antibiotic use, its impact on the microbial community, and a limitation of this study (Lines 463–467).

- “Co-detected viruses could represent either active replication of an additional virus (either newly acquired or reactivated) or leftover nucleic acid from a previous infection; the additional virus may be either as a non-pathogenic ‘bystander’ or a true disease-causing agent.” – suggest rewriting for clarity

Thank you for your comment. We have revised the sentence to improve clarity. This sentence now reads “Co-detected viruses may indicate either residual nucleic acid from a previous infection or active replication of an additional virus (newly acquired or reactivated). This additional virus could be either a non-pathogenic ‘bystander’ or a true disease-causing agent.” (Lines 372–375)

- “Although not definite, unique read count as a surrogate for viral load may provide a clue to differentiate between these possibilities. Co-detected viruses with low viral load may come from prolonged shedding from a previous infection, whereas those with high viral load are more likely to cause symptoms. In this study, we focused on viruses with evidence of substantial viral load based on read numbers, as being consistent with active replication and therefore possible co-infection.” Would like to see citations for this paragraph.

Thank you for the suggestion. We have included citations here (Line 378).

Overall, this is a well-structured paper. A few minor edits will improve readability. The discussion lacks potential explanation for the phylogenetic analysis and subsequent result in any detail. The supplemental phylogenetic analysis indicates a large analysis was performed. The bioinformatic methods section was well written and easy to follow, which will be appreciated by anyone replicating or applying it.

Thank you very much for the positive feedback and comments. The phylogenetic analyses of RSV and the results have been previously published; we have added the reference to our previous publication on the phylogenetic analyses (Lines 181–183). Regarding the phylogenetic analyses of enterovirus and human herpesvirus 6, the aim was to show that we could conduct phylogenetic inference using the reconstructed sequences from our sequencing method. No additional analysis was made.

REVIEWERS' COMMENTS

Reviewer #1 (Remarks to the Author):

Thank you for addressing all of our comments and requests for clarification. After review of the revised manuscript, we just have one additional question.

In the cohort of patients with Haemophilus, was the administration of antibiotics associated with improved outcomes?

Reviewer #2 (Remarks to the Author):

Reviewer #3 (Remarks to the Author):

The revised manuscript by Lin et al has been improved by addressing most of the reviewers' comments and I acknowledge that the authors have specifically addressed my questions and comments. I think the discussion could be tightened and shortened as it feels somewhat long as it stands.

I spotted a minor typo in the supplementary document, page 36 "respectively".

Reviewer #3 (Remarks on code availability):

I checked that the code exists and that it looks reasonable as well as that the README file exists and that the input data file is there too. I did not attempt to run it.

Reviewer #4 (Remarks to the Author):

I appreciate the authors' response to initial reviewer feedback and their incorporation of

suggested changes. There is a clear improvement in the narrative flow and the discussion addresses each result presented in this study. As mentioned before the detail in the methods section regarding the bioinformatics workflow will lend itself to reproducibility and will be appreciated by anyone following its logic and steps. I would recommend this study for publication.

RESPONSE TO REVIEWER COMMENTS

We express our gratitude to all the reviewers for reviewing our manuscript and point-by-point responses. We also appreciate their positive opinion and additional comments. Please see below our point-by-point responses to the additional comments. All line numbers included in the responses refer to the manuscript with tracked changes.

Reviewer #1 (Remarks to the Author):

Thank you for addressing all of our comments and requests for clarification. After review of the revised manuscript, we just have one additional question.

In the cohort of patients with *Haemophilus*, was the administration of antibiotics associated with improved outcomes?

Thank you for the question. In the cohort of infants with *Haemophilus*, antibiotic administration was associated with more severe disease presentation, including higher ReSVinet scores and an increased need for hospitalisation, intensive care, respiratory support, and mechanical ventilation. The only outcome variable not associated with antibiotic administration was fever. These results have been added to the main text (Lines 289– 291) and Supplementary Table 17.

We also conducted an additional analysis on the association between disease severity and the presence of *Haemophilus* spp., including antibiotic administration as one of the covariates. We were able to confirm that the positive association between disease severity and *Haemophilus* co-detection was robust to antibiotic treatment (Lines 291– 293).

Reviewer #2 (Remarks to the Author):

We are grateful to the reviewer for their support in the peer review of our work.

Reviewer #3 (Remarks to the Author):

The revised manuscript by Lin et al has been improved by addressing most of the reviewers' comments and I acknowledge that the authors have specifically addressed my questions and comments. I think the discussion could be tightened and shortened as it feels somewhat long as it stands.

I spotted a minor typo in the supplementary document, page 36 "respectively".

Thank you for the comments. We have shortened the discussion by 17%, reducing the word count from 1,675 to 1,382. The typo in Supplementary Fig. 2 has also been corrected.

Reviewer #4 (Remarks to the Author):

I appreciate the authors' response to initial reviewer feedback and their incorporation of suggested changes. There is a clear improvement in the narrative flow and the discussion addresses each result presented in this study. As mentioned before the detail in the methods section regarding the bioinformatics workflow will lend itself to reproducibility and will be appreciated by anyone following its logic and steps. I would recommend this study for publication.

Thank you for your positive feedback and recommendation.